# Rotary-wing drone-induced flow – comparison of simulations with lidar measurements

Liqin Jin[1, *], Mauro Ghirardelli[2, *], Jakob Mann[1], Mikael Sjöholm[1], Stephan T. Kral[3], and Joachim Reuder[2, 3]

[1]Department of Wind and Energy Systems, Technical University of Denmark, Frederiksborgvej 399, 4000 Roskilde, Denmark.
[2]Geophysical Institute and Bergen Offshore Wind Centre, University of Bergen, 5020 Bergen, Norway
[3]Geophysical Institute and Bjerknes Centre for Climate Research, University of Bergen, 5020 Bergen, Norway
[*]These authors contributed equally to this work.

**Correspondence:** Mauro Ghirardelli (mauro.ghirdelli@uib.no), Liqin Jin (liqn@dtu.dk)

**Abstract.** Ultrasonic anemometers mounted on rotary-wing drones have the potential to provide a cost-efficient alternative to the classical meteorological mast-mounted counterpart for atmospheric boundary layer research. However, the propeller-induced flow may deteriorate the accuracy of free-stream wind velocity measurements by wind sensors mounted on drones, which needs to be investigated for optimal sensor placement. Computational fluid dynamics (CFD) simulations are an alternative to experiments for studying characteristics of the propeller-induced flow but require validation. Therefore, we performed an experiment using three short-range continuous-wave Doppler lidars (DTU WindScanners) to measure the complex and turbulent three-dimensional wind field around a hovering drone at low ambient wind speeds. A good agreement is found between experimental results and those obtained by CFD simulations under similar conditions. Both methods conclude that the disturbance zone (defined by a relative deviation from the mean free-stream velocity by more than $1\%$) on a horizontal plane located at $1\,D$ (rotor diameter $D$ of $0.71\,\mathrm{m}$) below the drone, extends about $2.8\,D$ upstream from the drone center for the horizontal wind velocity, and more than $7\,D$ for the vertical wind velocity. By comparing wind velocities along horizontal lines in the upstream direction, we find that the velocity difference between the two methods is $\leq 0.1\,\mathrm{m\,s^{-1}}$ (less than $4\%$ difference relative to the free-stream velocity) in most cases. Both the plane and line scan results validate the reliability of the simulations. Furthermore, simulations of flow patterns in a vertical plane at ambient speed of $1.30\,\mathrm{m\,s^{-1}}$ indicate that it is difficult to accurately measure the vertical wind component with less than $1\%$ distortion by drone-mounted sonic anemometers.

## 1 Introduction

The proper characterization of atmospheric flow velocities and turbulence is essential to understanding the structure and dynamics of the atmospheric boundary layer (ABL) (Stull, 1988; Wyngaard, 2010). It is, therefore, crucial to obtain accurate wind and temperature measurements with high spatial and temporal resolutions for a variety of basic and applied ABL research topics, such as weather and climate prediction (Teixeira et al., 2008), wind energy meteorology (Emeis, 2010; Albornoz et al., 2022), and atmospheric modeling (Etling, 1996). Historically, sonic anemometers have been the most common instru-

ment for measuring atmospheric flow and turbulence since they were introduced in the 1950s (Suomi, 1957). Compared to cup anemometers, which measure only the magnitude of the horizontal wind vector, sonic anemometers can determine all three components of the turbulent wind velocity with high accuracy (MacCready, 1966; Izumi and Barad, 1970), by measur-

ing the ultrasonic waves' flight time along a path between two transducers (Kaimal et al., 1968). Additionally, compared to Doppler wind lidars (LIght Detection and Ranging), sonic anemometers have a smaller measurement volume, making them more suitable for studying turbulent fluctuations of higher frequency (Held and Mann, 2018).

Traditionally, sonic anemometers are mounted on meteorological masts (met masts), providing single-point measurements. Even though most of our current understanding of atmospheric turbulence comes from measurements performed with mast-

mounted sonic anemometers, masts or towers as sensor carriers limit the measurement flexibility considerably. Aside from this, mast-mounted sonic anemometers may suffer from flow distortion caused by the tower itself (Dyer, 1981; McCaffrey et al., 2017), deteriorating measurement accuracy. Consequently, sonic anemometers can underestimate wind velocity and overestimate turbulent fluctuation if they measure the deficit velocity in the wake of support structures. These limitations necessitate the advancement of measurement techniques beyond the traditional mast-based approach.

Since the beginning of the 21st century, uncrewed aerial vehicles (UAVs) with rotary-wing have become more popular for conducting atmospheric measurements (Hemingway et al., 2017; Leuenberger et al., 2020; Tikhomirov et al., 2021), due to their flexibility in orienting, precise hovering capabilities and ease of deployment. Wind velocity and direction can be reconstructed from either the avionic information of UAVs alone (Neumann and Bartholmai, 2015; Palomaki et al., 2017; Segales et al., 2020; Wetz et al., 2021; González-Rocha et al., 2023), or from wind sensors mounted on the UAVs. Even

though the former indirect approach is well-established and has the advantage of not requiring external measurement devices, it has limitations in resolving three-dimensional wind fields and fine-scale turbulent fluctuations. An exception is found in Wildmann and Wetz (2022), where all three velocity components are measured at a frequency of 1 Hz. However, a sampling frequency of 10 Hz to 20 Hz is typically necessary to resolve the smallest turbulent scales. Therefore, the latter method of using fast-response 3D anemometers, such as sonic anemometers, is inevitable for direct observations of turbulence. This approach

extends measurement capabilities, but it may reduce flight performance due to the added weight.

In the past few years, new, compact, and lightweight anemometers have been developed and used on rotary-wing UAVs, mainly to measure mean wind speeds in the horizontal direction or vertical ABL wind profiles (Palomaki et al., 2017; Shimura et al., 2018; Li et al., 2023). Compared to their full-size counterparts, these smaller sensors may have limitations in sampling capabilities, especially in measuring vertical velocity. However, the potential of UAVs equipped with full-size sonic anemometers

has not been extensively explored, due to size and weight constraints as well as stability challenges associated with mounting a heavy payload (Natalie and Jacob, 2019). Despite these challenges, we hypothesize that the integration of full-size sonic anemometers on UAVs will enable more accurate and comprehensive atmospheric data collection.

The study by Thielicke et al. (2021) highlighted the importance of evaluating the propeller-induced flow (PIF) when mounting a full-size sonic anemometer on top of a quad-copter with considerable commitment in infrastructure (use of wind tunnel)

and time (multiple calibration flights). Essentially, the study concluded that by mounting the wind sensor away from the drone's fuselage, the influence of PIF on the measurements can be effectively reduced. This strategy was followed also by Wilson et al.

(2022), where they suggested that a vertical separation distance of $5.3$ rotor diameters ($5.3\,D$) should be sufficient to mini-mize PIF when placing the sonic anemometer centered above the drone. Vasiljević et al. (2020) presented a proof-of-concept drone–lidar system and concluded that the lidar should be placed out of the drone's disturbance zone stretching between 1 and

$2\,\text{m}$ ($1.9\,D$ and $3.7\,D$ with rotor diameters of $0.53\,\text{m}$) from the center, based on measurements of radial wind speed.

Although moving the sensor away from the fuselage seems like a straightforward solution to reduce PIF interference, it introduces additional complexity to the system. A drone's PIF varies in features since it depends on both internal and external factors, such as its architecture and design (Guillermo et al., 2018; Lei and Cheng, 2020; Lei et al., 2020), as well as the presence of walls, altitude above the ground, and wind conditions (Zheng et al., 2018; Lei and Lin, 2019; Guo et al., 2020).

Apart from this, mounting additional weight away from the drone's center of gravity can impair its flight stability. Therefore, it is necessary to assess each drone and wind sensor combination individually.

A computational fluid dynamics (CFD) study is a feasible and efficient alternative to field experiments and wind tunnel tests (Schiano et al., 2014), and has gained considerable popularity in UAV research (Paz et al., 2021) due to its capability to analyze propeller performance as well as to assess workloads (Deters et al., 2014; Kutty and Rajendran, 2017). The majority

of studies have primarily examined how the PIF impacts the drones' flight stability for design purposes (Zheng et al., 2018; Guillermo et al., 2018; Lei and Lin, 2019; Guo et al., 2020), while our research focuses on optimizing drone-mounted sensor placement ensuring a minimal PIF influence. We propose a new design with a boom-mounted sonic anemometer that faces upwind and is placed below the fuselage of a rotary-wing drone. In this arrangement, the drone's main batteries can be placed on the boom's opposite side as a counterbalance. Furthermore, increasing the battery capacity would extend the flight time

and shift the battery position closer to the fuselage center. CFD simulations of this design were presented in Ghirardelli et al. (2023), which were computed efficiently by simplifying the drone's geometry and setting up ideal flow conditions. However, this simulation model must be experimentally validated before its results can be used as a reference for sensor placement.

Apart from simulations, high-resolution lidar remote sensing is a promising approach to studying the complex PIF. Continuous-wave (CW) Doppler lidar can remotely obtain accurate three-dimensional flow observations without disturbing the flow. Con-

sequently, CW lidars are extensively applied to detect wind profiles (Köpp et al., 1984; Peña et al., 2009); assess wind resources (Bingöl et al., 2009; Viselli et al., 2019); test wind turbines' performance based on wake measurements (Wagner et al., 2014; Shin and Ko, 2019; Fan et al., 2023); predict the incoming gusts and flow to reduce loads (Bos et al., 2016); study turbulence around a suspension bridge (Cheynet et al., 2016); and in the near-wake region of a tree (Angelou et al., 2022), with good spatial and temporal resolutions. Recently, two CW lidars were applied to measure the two-dimensional downwash wind fields

in a horizontal and a vertical plane below a hovering search and rescue helicopter (Sjöholm et al., 2014).

Being aware of the capability of lidar measurements, we conducted a field measurement campaign using three synchronized CW Doppler lidars to reconstruct the three-dimensional flow field around and below a drone. The goal is to validate CFD simulations based on the setup presented by Ghirardelli et al. (2023). Such a CFD setup can be further used to determine the optimal placement location for a sonic anemometer on a large multi-copter drone (diameter of $1.88\,\text{m}$). To the best of the

authors' knowledge, this is the first study to use three CW lidars to investigate the turbulent three-dimensional flow around a rotary-wing drone.

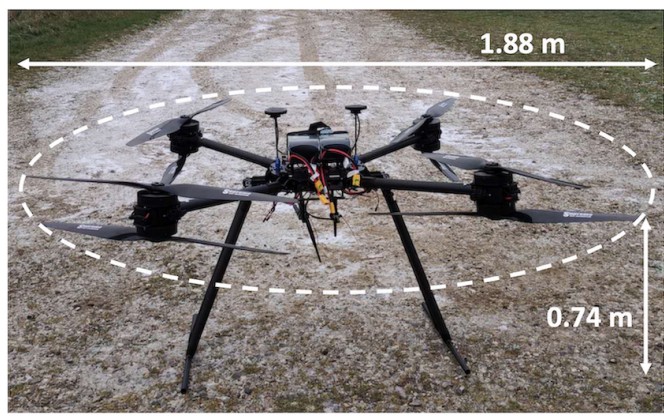

**Figure 1.** The Foxtech D130 x8, rotary-wing drone used in the experiment.

In Section 2, the instruments and CFD simulations employed are elaborately described. Section 3 introduces the field measurement campaign and the wind characteristics obtained by a tower-mounted sonic anemometer nearby. The principle of Doppler spectral processing to retrieve wind vectors is presented in detail in Section 4 and the comparison of wind fields retrieved by the lidar measurements and CFD simulations is shown in Section 5. The most important findings of our study are summarized in the Discussions and conclusions (Section 6).

## 2   Instrumentation and model

### 2.1   The rotary-wing drone

The drone utilized in this study is the Foxtech D130 x8, a rotary-wing drone equipped with eight propellers arranged in four pairs of contra-rotating open rotors (Fig. 1). Each rotor has a diameter of $0.71\,\mathrm{m}$.

Each pair consists of two propellers spinning in opposite directions, driven by brushless electric motors (T-motor U10II KV100). The drone's take-off weight reaches $13.5\,\mathrm{kg}$ and it has a nominal maximum flight time of $45\,\mathrm{min}$. The system incorporates a Cube Orange autopilot unit that is connected to two Here3 GNSS antennas, which enables real-time kinematic navigation capabilities when paired with a Here+ GNSS base station. The autopilot operates under the open-source ArduCopter flight controller and provides position and attitude data with a sampling frequency of $8\,\mathrm{Hz}$.

### 2.2   CFD simulations setup

A total of seven CFD simulations were performed using Ansys Fluent 2022 R1. The simulation design relies on the study by Ghirardelli et al. (2023), where it is extensively described. The inflow wind speeds for this comparison were carefully adjusted to match the field experiment. The actual geometry of the drone is simplified according to the actuator disc theory (Rankine, 1865; Froude, 1889; Sayigh, 2012). The drone is treated as eight two-dimensional discs (Fig. 2a) that apply an instantaneous

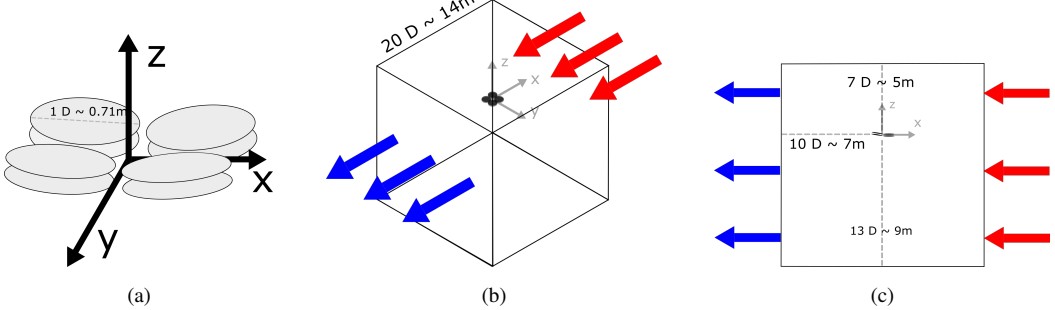

(a)                                       (b)                                       (c)

**Figure 2.** Schematic diagram of the computational domain with the flow entering through the inlet (red arrows), exiting through the outlet (blue arrows), and the remaining external faces considered as walls. **(a)** Eight actuator discs. **(b)** A three-dimensional isometric projection of the domain. **(c)** Side view.

pressure jump in the flow, and that are placed along the planes of rotation of the real drone propellers ($0.71\,\mathrm{m}$ diameter). Disregarding the intricate details of each rotor blade's geometry has the advantage of reducing the real geometry mapping process and the computational time. At the geometrical center of mass resulting from the eight rotor discs, there is a global reference point $(0, 0, 0)$.

To compensate for the difference between the average drone heading and the wind direction observed during the individual validation flights, the geometry is rotated around the vertical axis (yaw). Furthermore, as the drone tilts, while hovering to compensate for the wind drag (Anderson, 2011), the mean rotor plane where the actuator discs lie has been tilted according to the average tilt over each measurement period. The actuator discs are enclosed within a cubic volume domain that is $20\,D$ wide and tall, laterally centered. Vertically, the center is located $7\,D$ from the top and $13\,D$ from the bottom. This disposition ensures that the influence between the propeller-induced flow and the bottom wall is reduced. The wind speed is kept constant over the inflow plane.

The computational grid for the CFD simulations was automatically generated using Ansys Fluent Meshing with the Watertight Geometry workflow. This method simplifies mesh generation for CFD simulations, allowing users to perform all stages of the simulation, including meshing and post-processing, within a single software session and user interface (Ansys Fluent, 2023). The grid consists of a poly-hex-core mesh (Zore et al., 2019) and it was the subject of a mesh refinement study in Ghirardelli et al. (2023). To ensure accuracy, the maximum cell size within the domain was set to $0.3\,\mathrm{m}$, while on the actuator discs, it was $0.02\,\mathrm{m}$ and overall, the grid comprises $1.66$ million cells. The k-$\epsilon$ turbulence closure model is commonly used in CFD simulations for turbulent flows, especially in free-shear layer scenarios. It assumes fully turbulent flow and does not consider molecular viscosity effects. The selection of the k-$\epsilon$ model, instead of the k-omega and k-omega SST models, was based on the study's emphasis on non-wall flow characteristics.

Table 1 summarizes the setup parameters for the geometry and flow velocities used in the CFD simulations. Following the initial mesh setup and selection of the turbulence model, we used standard settings from Ansys Fluent to ensure consistency and reliability.

**Table 1.** Summary of setup parameters of the CFD simulations and relative heights of the lidar scans. Flow above the drone is signified by positive $\Delta h$ and vice versa flow below the drone by negative $\Delta h$. The yaw and tilt positions are also input to the simulation.

| Measurement | Inlet Velocity $U_0$ ($\mathrm{m\,s^{-1}}$) | $\Delta h$ (m) | $\Delta h/D$ | Yaw Angle (°) | Tilt Angle (°) |
|---|---|---|---|---|---|
| Plane Scans: | 4.09 | −0.7 | −1 | 21.2 | 2.7 |
| | 3.53 | −2.1 | −3 | 26.4 | 2.8 |
| | 3.54 | −3.7 | −5.2 | 14.0 | 2.9 |
| | 3.92 | −4.5 | −6.3 | 34.4 | 3.3 |
| Line Scans: | 1.34 | 3.0 | 4.2 | 174.3 | 1.5 |
| | 1.51 | 2.2 | 3.1 | 174.2 | 1.6 |
| | 1.77 | −1.6 | −2.3 | 354.3 | 1.4 |

## 2.3 The WindScanner system

The ground-based, short-range WindScanner system developed by DTU Wind and Energy Systems consists of three synchronized coherent CW Doppler lidars (Fig. 3), which are capable of accurately retrieving wind vectors and measuring turbulence (Sjöholm et al., 2009; Mikkelsen et al., 2020; Jin et al., 2023). A single CW Doppler lidar can only measure the one-dimensional projection $v_{LOS}$ of wind velocity vector along its line-of-sight beam direction. By combining the independent and simultaneous measurements of $v_{LOS}$ from three Doppler lidars, a full three-dimensional wind vector can be retrieved.

The use of CW Doppler lidars is beneficial for a variety of wind energy applications. Despite this, CW Doppler lidars are susceptible to moving objects away from the intended focus point, such as flying birds. Besides, their spatial resolution decreases as the focus distance increases, which may deteriorate the accuracy of wind velocity and turbulence measurements by CW lidars (Jin et al., 2022b). Therefore, we placed the three lidars as close to the intended scanning positions as possible to compact the measurement volume (Angelou et al., 2012) and minimize potential biases resulting from volume-averaging

(Clive, 2008; Sjöholm et al., 2009; Forsting et al., 2017). To improve the accuracy of the flow velocity retrieval, we discard spectra containing Doppler shifts caused by hard targets and out-of-focus moving objects during the post-processing (Jin et al., 2022a).

  The detected backscatter signal by the lidars is mixed with the local oscillator and sampled at $120\,\mathrm{MHz}$ and the Fast Fourier Transform (FFT) frequency resolution becomes $234.4\,\mathrm{kHz} = (120\,\mathrm{MHz})/512$ when Doppler spectra are calculated with 512

frequency bins. Consequently, the line-of-sight velocity bin resolution is $0.183\,\mathrm{m\,s^{-1}} = (1.565\,\mathrm{\mu m}/2)\cdot(234.4\,\mathrm{kHz})$, which is calculated by the FFT frequency resolution and the laser wavelength $\lambda$. After a block averaging of 726 spectra to reduce noise fluctuations, the final spectrum is sampled at a frequency of $322\,\mathrm{Hz} = (120\,\mathrm{MHz})/(512\cdot726)$ with the corresponding sample time of 3.1 milliseconds for each spectrum. Therefore, each lidar provides a data file with 19320 spectra for every minute of measurement. In addition, to distinguish the blue or red Doppler shift depending on whether the aerosols move towards or

away from the lidar, the in-phase/quadrature-phase (IQ) homodyne detection method (Abari et al., 2014) is employed.

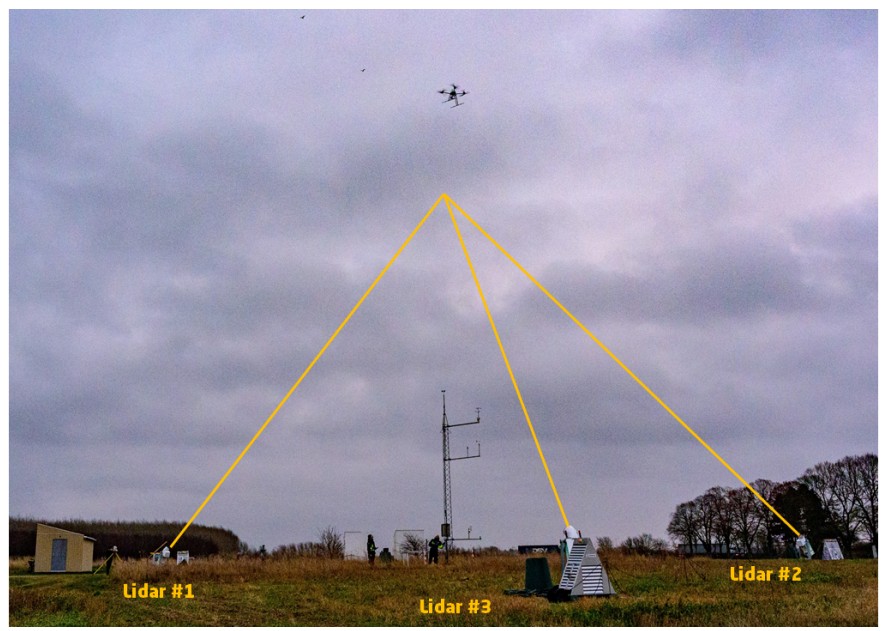

**Figure 3.** Experiment setup of the three CW Doppler lidars at DTU Risø campus. The surrounding terrain is relatively flat and agricultural. The three lidars focus at about $8\,\text{m}$ above the top of a met mast.

## 3 Experimental setup

On the 14th and 15th of December 2022, we performed a field experiment with the WindScanner system to scan the PIF above and below the hovering drone at Risø campus of the Technical University of Denmark (DTU), as depicted in Fig. 3. Considering the lidars' shortest focus distance and a safe height from the drone to the reference met mast, the line-of-sight focus distance of the three lidars varied between $24.8\,\text{m}$ and $30.9\,\text{m}$, with the corresponding elevation angle ranging from $42.3°$ to $27.8°$. The probe length of each lidar, or more precisely the full-width-at-half-maximum (FWHM) of the Lorentzian-shaped weighting function, was in the range of $0.56\,\text{m}$ and $0.87\,\text{m}$, which was calculated by

$$\text{FWHM} = 2 \cdot z_R = 2 \cdot \frac{\lambda \cdot R^2}{\pi a_0^2} \tag{1}$$

where $z_R$ is the Rayleigh length, $\lambda = 1.565\,\mu\text{m}$ the laser wavelength, $R$ the distance from the lidar to where the beam is focused, and $a_0 = 33\,\text{mm}$ is the effective beam radius for the 6-inch lidar lens that used in this study.

After focus calibration with a rotating hard target, the three lidars were programmed to scan synchronously a horizontal plane and a horizontal line, both centered above the met mast with a height distance of eight meters above its top. The flow-field evolution in a horizontal plane at a certain distance below the drone was measured for comparison with the CFD simulations, while fast line scans enable detailed comparisons in the upstream direction, which is the most promising region for a boom-mounted sonic anemometer attached to the drone Ghirardelli et al. (2023). For the measurements, we defined a right-handed

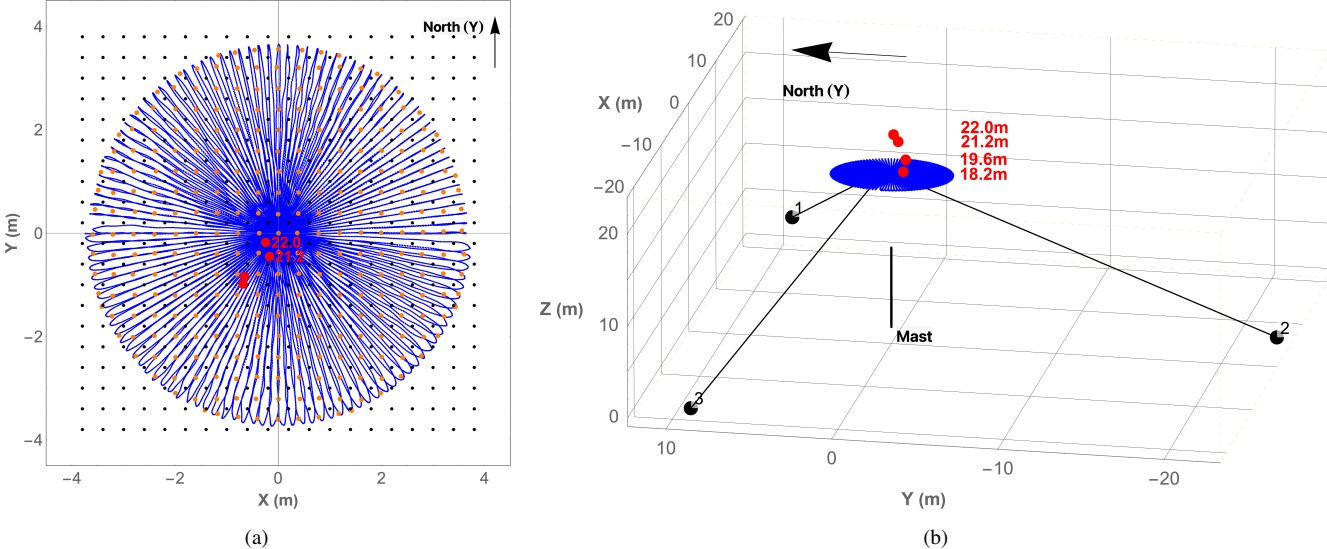

**Figure 4.** The experimental setup for the plane scans. **(a)** Top-down view along the negative $z$-axis. **(b)** Side view. The three lidars are marked by numbers 1, 2, and 3, while the solid black line indicates the met mast. The scanning plane is at $z = 17.5\,\text{m}$ throughout the measurement campaign and the drone's average center positions and heights are indicated by the red dots. The black dots in **(a)** are the centers of the grid cells for grouping the scanning points, while the orange dots are the average measurement points in each cell.

coordinate system with the origin located close to the bottom of the met mast, the $y$-axis pointing towards the geographic north, the $x$-axis pointing east, and $z$ up (Fig. 4 and Fig. 5).

The horizontal disc with a diameter of $7.4\,\text{m}$ ($10.4\,D$) was scanned using a trajectory of 60 lines. The three laser beams followed a path that started from one side of the line, passed through the disc center to the other end of the line, and then

returned over the same route to its starting point. This allows the dual-prism lidars to complete the pattern in a reasonably short time if the symmetry axis of each lidar crosses the disc center and the trajectory in the vicinity passes exactly through the same crossing point. The scanning duration of each horizontal line was $1\,\text{s}$, including the transition time to rotate $3°$ around the center of the line to the adjacent line. Therefore, the completion of the whole scanning plane took one minute.

During the horizontal plane scans lasting from 13:35 to 13:57 local time (all times mentioned in the paper are UTC+1), the

drone hovered at four heights and stayed for five minutes at each height. The corresponding height difference $\Delta h$ between the drone plane and the scanning plane is $\Delta h = -0.7\,\text{m},\ -2.1\,\text{m},\ -3.7\,\text{m}$ and $-4.5\,\text{m}$ ($-1\,D,\ -3\,D,\ -5.2\,D$, and $-6.3\,D$ respectively). The negative sign indicates that the scanning plane is below the drone. Fig. 4 shows the average drone positions for the four heights (red points). During the lidar measurement, the drone drifted slightly southwest of the plane center because the Real Time Kinematics system was not properly configured, despite the intention to hover the drone at the center.

Similarly to the plane pattern, the cycle duration of the line scan was $1\,\text{s}$. In an attempt to align with the wind direction, the line spanned from $(x, y) = (-0.32, -3.62)\,\text{m}$ to $(+0.34, +3.64)\,\text{m}$. Thus, it pointed approximately $5°$ anticlockwise from the north. This resulted in completing approximately 60 iterations of the line scan per minute. Line scans were performed by

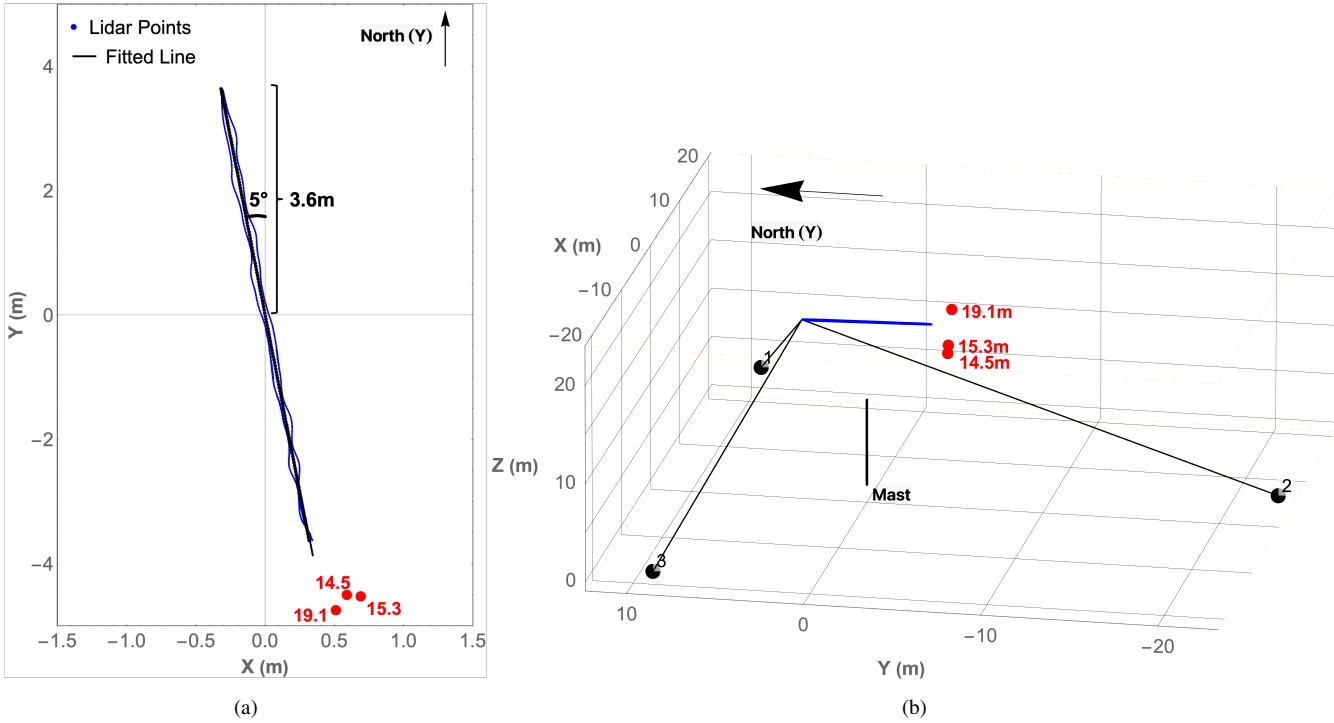

**Figure 5.** The experimental setup for the line scans. **(a)** Top-down view along the negative $z$-axis. **(b)** Side view. The three lidars are marked by numbers 1, 2, and 3, while the solid black line in **(b)** indicates the met mast. The measurements were taken at a height of $17.5\,\mathrm{m}$, the same as the plane scan. The black line in **(a)** is the fitted scanning line.

hovering the drone at seven different heights, either at the end or side of the line. At each height, the drone stayed for five minutes. However, only the three heights close to the line's end were considered to analyze the upstream flow, as shown in Fig. 5. Therefore, the height difference $\Delta h$ is 3.0 ($4.2\,D$), 2.2 ($3.1\,D$), and $-1.6\,\mathrm{m}$ ($-2.3D$). The corresponding orthogonal distance in the horizontal direction from the drone position to the scanning line in Fig. 5a is $0.2\,\mathrm{m}$, $0.29\,\mathrm{m}$ and $0.1\,\mathrm{m}$.

The 10-min wind characteristics measured by sonic and cup anemometers $18\,\mathrm{m}$ a.g.l. on the met mast west of the DTU V52 wind turbine ($358\,\mathrm{m}$ north to the met mast in Fig. 3) are presented in Fig. 6. Sonic anemometer measurements indicated that the average horizontal wind velocity for the plane scans was $3.44\,\mathrm{m\,s^{-1}}$, while it was $2.11\,\mathrm{m\,s^{-1}}$ for the line scans. The 10-min wind direction was from $-22.7°$ (northwest) during the plane scans and later changed to $9.95°$ (northeast) for the line scans. Additionally, the 10-min vertical wind component varied from $-0.13\,\mathrm{m\,s^{-1}}$ to $-0.17\,\mathrm{m\,s^{-1}}$ during the measurement period. As shown in Fig. 6, the average wind speed and direction obtained from the three lidars far from the drone are represented by green dots. Even if there are some differences, it is a good comparison since the lidars and the sonic anemometer are relatively far apart ($358\,\mathrm{m}$).

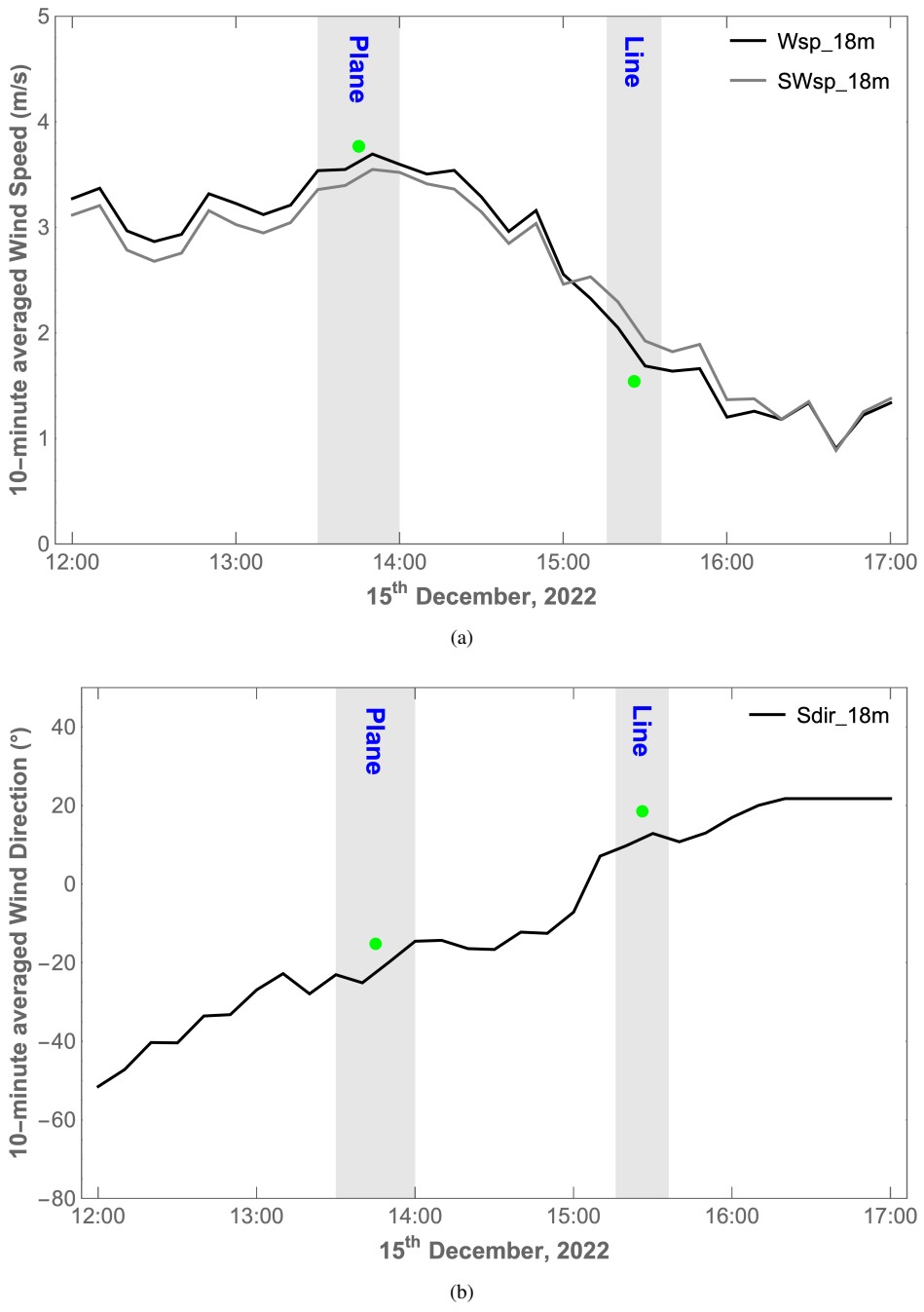

**Figure 6.** 10-min wind measurements by sonic and cup anemometers at 18 m height. **(a)** Wind speed measured by the sonic (SWsp) and cup (Wsp) anemometers. **(b)** Wind direction measured by the sonic anemometer (Sdir). The gray bars mark the plane scan period from 13:35 to 13:57 (UTC+1) and the line scan period from 15:16 to 15:36 (UTC+1). The green dots are the corresponding measurements by the lidars.

## 4 Post-processing and data analysis

A relatively stable estimate of the flow around the drone could be made after $3\,\mathrm{min}$ corresponding to 3 iterations of the plane scanning. The data acquired for the plane scan was grouped by the index of every square grid cell, which has a dimension of $0.4\,\mathrm{m} \times 0.4\,\mathrm{m}$ (the black dots in Fig. 4a). Hence, with a scanning step of $47\,\mathrm{mm}$ and a spectral sampling frequency of $322\,\mathrm{Hz}$ from the lidars, at least 50 Doppler spectra were present in each grid cell. Analysis of the line scan was performed using 3-min lidar data consisting of 180 iterations of the line cycle. We segmented the line every $50\,\mathrm{mm}$, resulting in at least 360 Doppler spectra per segment.

After the Doppler spectra were processed with the steps (Jin et al., 2023) of dividing raw spectra by the background noise, subtracting a spectral threshold, and replacing negative values with zeros, they were averaged in the same grid cell or segment. Thereafter, we applied the median method (the median of the accumulated energy in the spectrum) to calculate the line-of-sight wind velocity (Held and Mann, 2018), and retrieved the wind vectors based on the line-of-sight velocities determined from the three lidars.

Despite placing the three lidars as close as possible to minimize the probe length, they could still hit the drone body or the rotational propellers during measurement. As demonstrated in Fig. 7b, the Doppler signal appearing at the center of the spectrum is caused by the drone body, while the signal on the left side marked by the red arrow is induced by the propeller. Compared with the normal Doppler signal caused by the aerosols in Fig. 7a, the energy of the two peaks in Fig. 7b is much higher because of the reflected light from the hard targets. By adding up the power spectral density we can easily identify an area for each lidar where beams hit either the drone body or its propellers, see Fig. 7c and d. With an increasing height difference $\Delta h$, the strong backscatter area moves away from the drone center, as seen from Fig. 7d. Consequently, we filter out the Doppler spectra whose sum exceeds a certain threshold to eliminate the detrimental effects of hard targets.

## 5 Results

### 5.1 The flow field in a horizontal plane below the drone

The 3-min-average downwash flow in a horizontal plane simulated by CFD and retrieved by the three lidars is exhibited in Fig. 8 while the drone was hovering at $0.7\,\mathrm{m}$ $(1\,D)$ above the scanning plane. The free-stream wind velocity was $4.09\,\mathrm{m\,s^{-1}}$ along the positive $x$-axis with a turbulence intensity (TI) of $0.04$ and the drone was yawed about $21.2°$ to the incoming wind. In general, the flow patterns predicted by the CFD simulations and the lidar measurements are consistent.

For a clearer illustration, we normalized the horizontal wind velocity $U_h = \sqrt{U^2 + V^2}$ by subtracting the free-stream wind velocity $U_0$. Fig. 8a and b clearly illustrate how a blockage effect in the upstream induction zone slows the free-stream wind and the accelerated flow beside the downstream drone wake. According to Fig. 8c and d, the flow transverse to the inflow wind diverges upstream of the drone and converges downstream. The vertical velocity $W$ in Fig. 8e and f has a peak value of $12\,\mathrm{m\,s^{-1}}$ in the CFD simulations and $5.4\,\mathrm{m\,s^{-1}}$ in the observations, respectively. It should be noted, however, that lidar measurements did not capture the very detailed flow feature due to the effects of unresolved atmospheric turbulence, the averaging effect

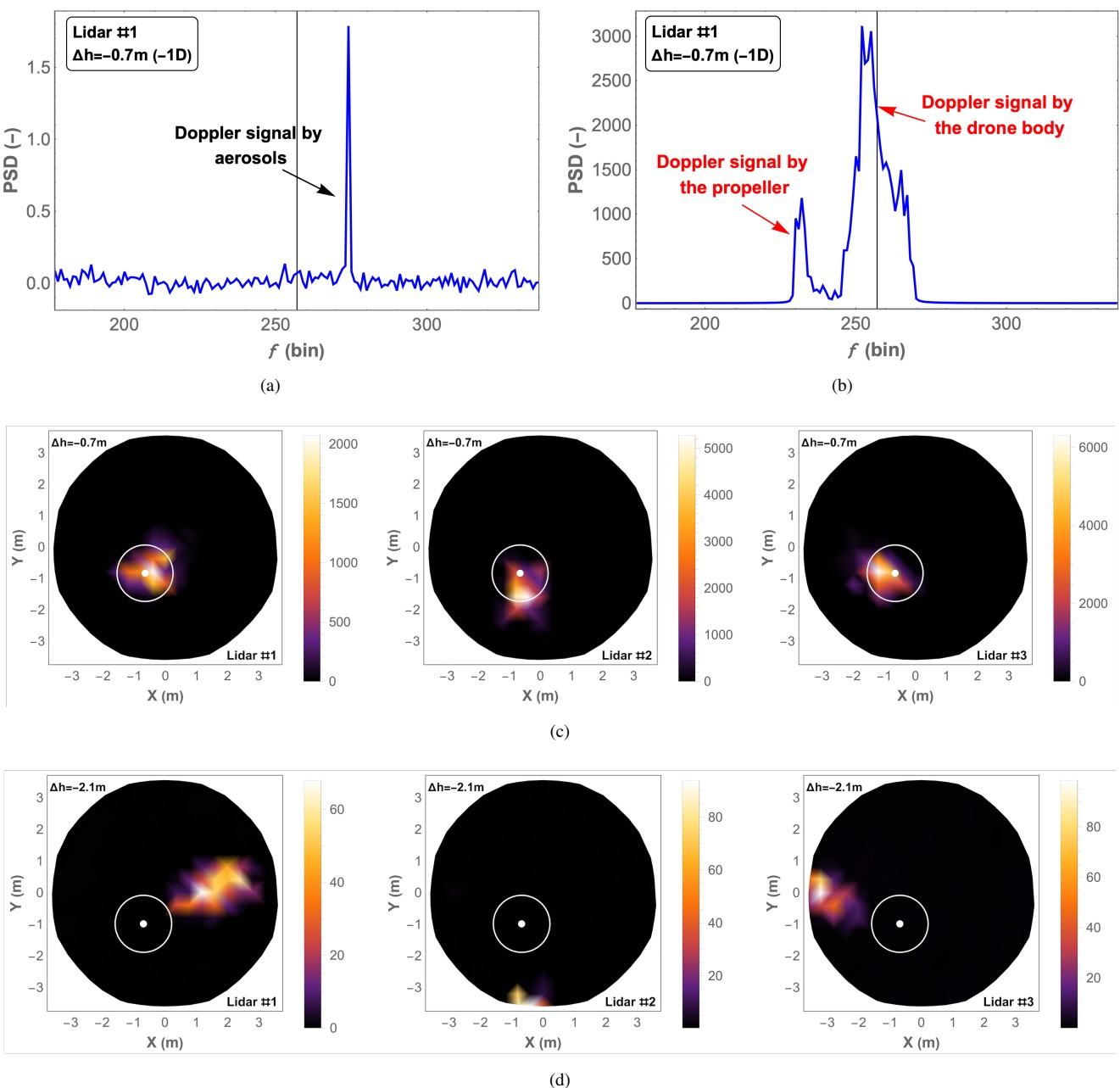

**Figure 7.** Examples of representative Doppler spectra and the sum of the power spectral density (PSD) within a reasonable frequency range where the Doppler signal occurs. **(a)** Noise-flattened spectrum containing only aerosol-induced Doppler signal. **(b)** Noise-flattened spectrum containing the Doppler signal caused by the drone. **(c)** Sum of PSD at $\Delta h = -0.7\,\mathrm{m}\,(-1\,D)$. **(d)** Sum of PSD at $\Delta h = -2.1\,\mathrm{m}\,(-3\,D)$. The white dot and circle are the drone center and diameter seen from above.

of lidars' measurement volume, and the drone drifting approximately $\pm 0.5\,\mathrm{m}$ along the free flow direction from its average position (the central black dot in Fig. 8).

Additionally, CFD simulations and lidar measurements indicate that the disturbance zone in horizontal wind velocity, defined as more than $1\%$ difference relative to the free-stream wind velocity, extends about $2\,\mathrm{m}$ $(2.8\,D)$ upstream from the drone. However, in terms of vertical wind velocity in Fig. 8e and f, it stretches more than $5\,\mathrm{m}$ $(7\,D)$.

A comparison of wind velocity obtained by the CFD simulations and lidar measurements is depicted in Fig. 9, where the drone was hovering at the three different heights displayed in Fig. 4. As the drone moves upwards increasing the distance to the scanning plane, both CFD simulations and lidar measurements show that the high-speed downwash is pushed downstream. When the drone was hovering at $2.1\,\mathrm{m}$ $(3\,D)$ above the plane, the accelerated area after the drone measured by the lidars was less prominent compared to the CFD simulations in Fig. 9c. This is probably due to the aforementioned effects of unsolved atmospheric turbulence, the averaging effect of lidars, and the drones' drifting. For the last two heights at $\Delta h = -3.7\,\mathrm{m}$ $(-5.2\,D)$ and $-4.5\,\mathrm{m}$ $(-6.3\,D)$ which is not shown, the drone's PIF has no significant impact in the studied area.

## 5.2 The flow field on horizontal lines around a drone

In addition to the measurement in a horizontal plane, the turbulent flow around the drone was also measured with upstream horizontal lines starting about $0.9\,\mathrm{m}$ upstream from the drone center in Fig. 10, while the drone was hovering at three heights (Fig. 5) and drifting around the average position within a radius of less than $0.25\,\mathrm{m}$. The averaged wind velocities over three minutes are presented in Fig. 11 and Fig. 12. A good agreement can be observed between the CFD simulations and lidar measurements for horizontal and vertical wind velocities, with differences of less than $0.1\,\mathrm{m\,s^{-1}}$ at all three heights studied, except for the vertical velocity $W$ at $\Delta h = -1.6\,\mathrm{m}$ $(-2.3\,D)$, which differs by $0.2\,\mathrm{m\,s^{-1}}$. At the three heights, the horizontal velocity difference $\Delta U_h$ between the two methods relative to $U_0$ is about $3.7\%$, $2.8\%$, and $5.2\%$ in Fig. 11. Particularly, for $\Delta h = 2.2\,\mathrm{m}$ $(3.1\,D)$, simulations fall within the uncertainty range (three times the standard error of the mean) of lidar measurements, as depicted in Fig. 11b and Fig. 12b.

Several factors contribute to the observed differences. Since the simulations are steady-state solutions based on the Reynolds-averaged Navier-Stokes (RANS) equations, they inherently lack detailed information about instantaneous flow fluctuations. Besides, the propellers are simplified based on the actuator disc theory, limiting the ability to accurately estimate the flow, especially when it comes to the turbulence generated by the real propellers. Additionally, the spatial averaging due to lidars' measurement volume and the temporal averaging of trajectory time series also contribute to the differences between observations and simulations.

The flow appears to be less turbulent as it approaches the drone, which is indicated by the shorter error bars in Fig. 11 and Fig. 12. This suggests that drone-mounted sonic sensors may impact the measurement of turbulence characteristics. However, this needs further investigation and is beyond the scope of the present study.

At all three heights, both methods yield about a nearly zero transverse velocity $V$ (not shown). It is also clearly demonstrated in Fig. 11 and Fig. 12 that there is a decrease in the horizontal wind velocity in the induction zone with $\Delta h < 0$, as well as an acceleration in the area above the drone with $\Delta h > 0$. The CFD simulations and lidar measurements agree well in the area

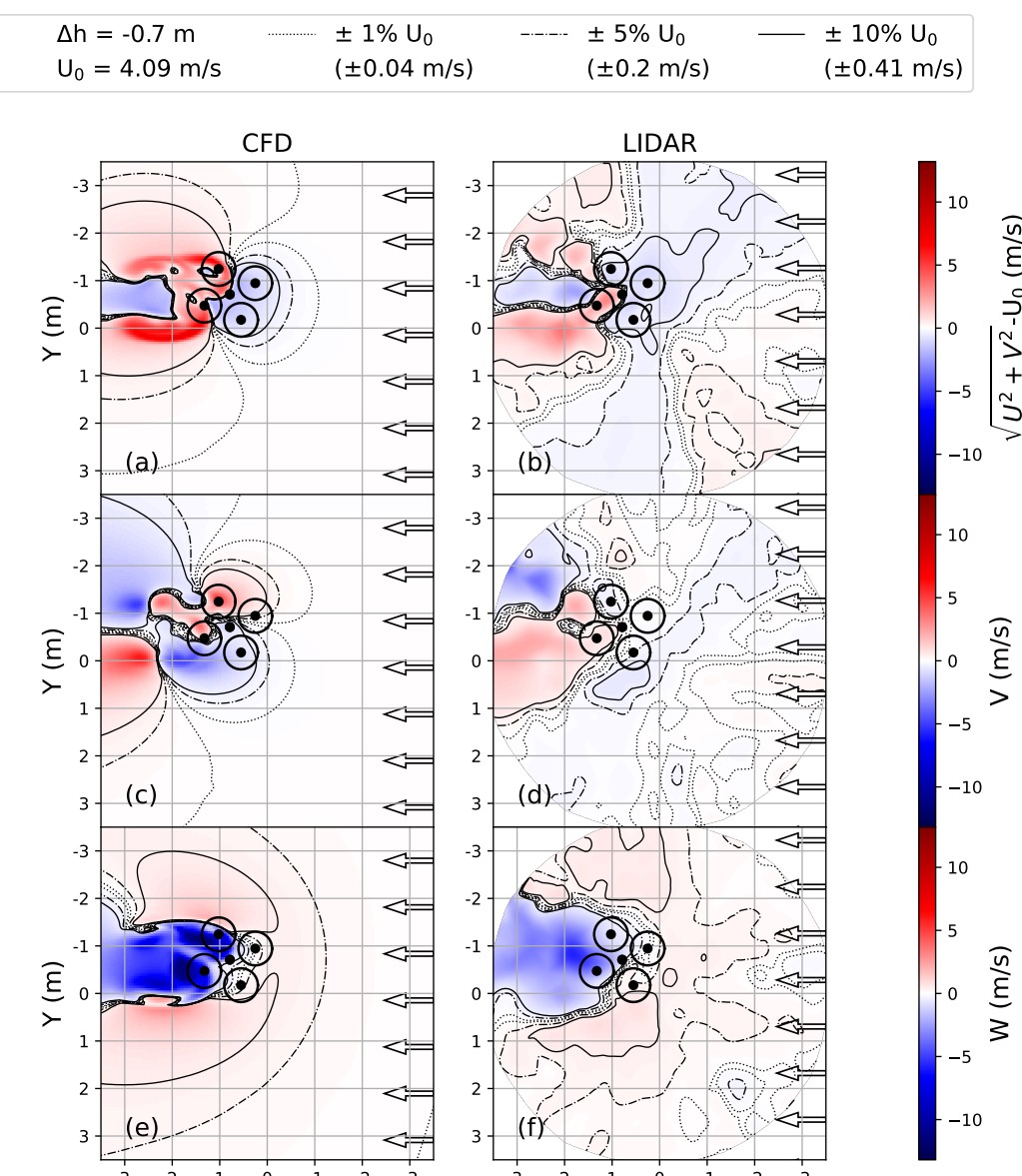

**Figure 8.** A comparison of wind fields and velocity contours of $1\%$, $5\%$, and $10\%$ deviation from $U_0$ obtained by CFD simulations (left column) and 3-min-average lidar data (right column) in a horizontal plane with the drone about $0.7\,\mathrm{m}$ ($1\,D$) above. Four circles represent the drone propellers and the black dot out of the circles is the average drone center.

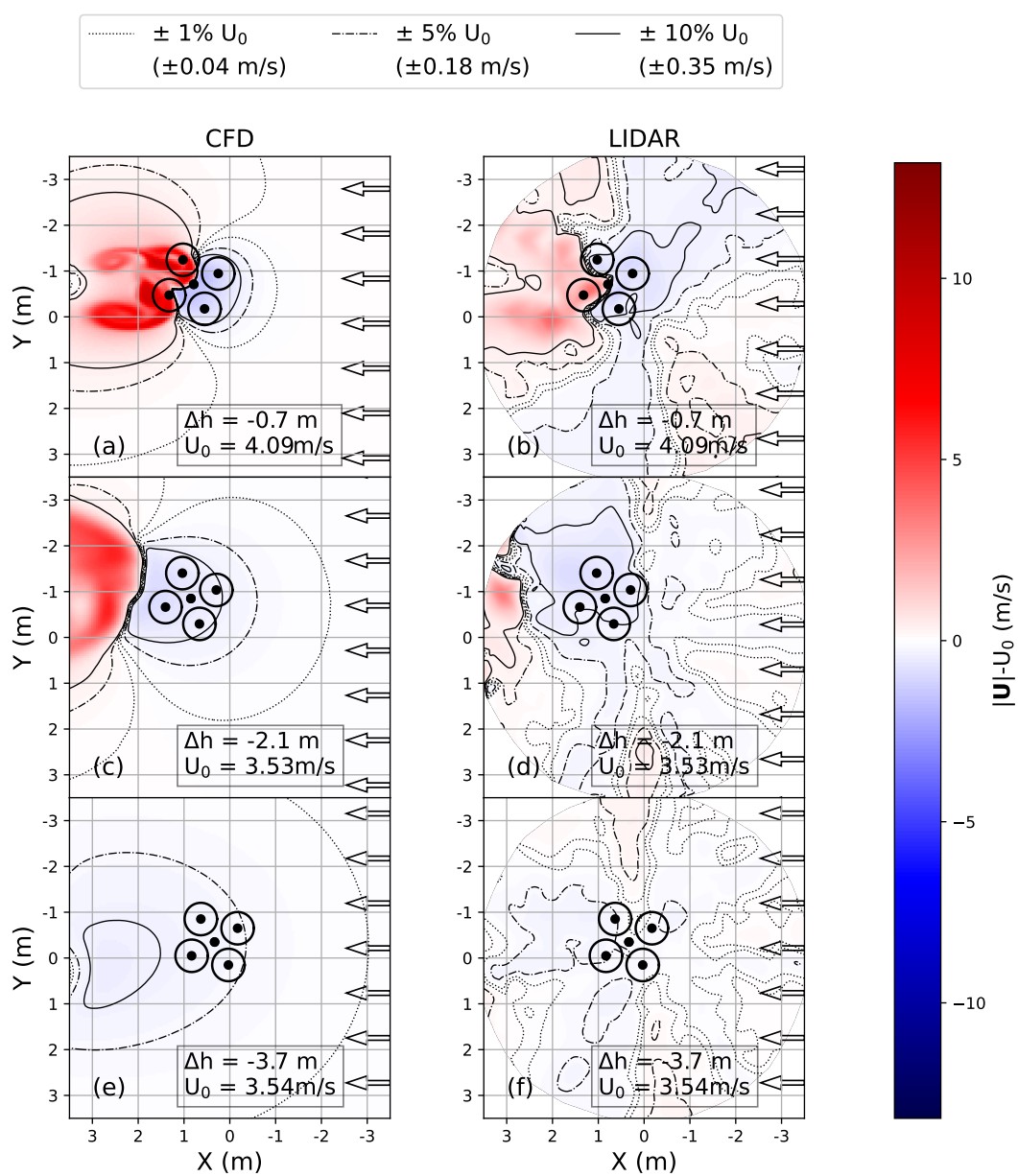

**Figure 9.** A comparison of wind velocity $|U|$ and velocity contours of $1\%$, $5\%$, and $10\%$ deviation from $U_0$ obtained by CFD simulations (left column) and 3-min-average lidar data (right column) in a horizontal plane, while the drone was hovering at $0.7\,\mathrm{m}$ ($1\,D$), $2.1\,\mathrm{m}$ ($3\,D$), and $3.7\,\mathrm{m}$ ($5.2\,D$) above the plane. Four circles represent the drone propellers and the black dot between the circles is the average drone center.

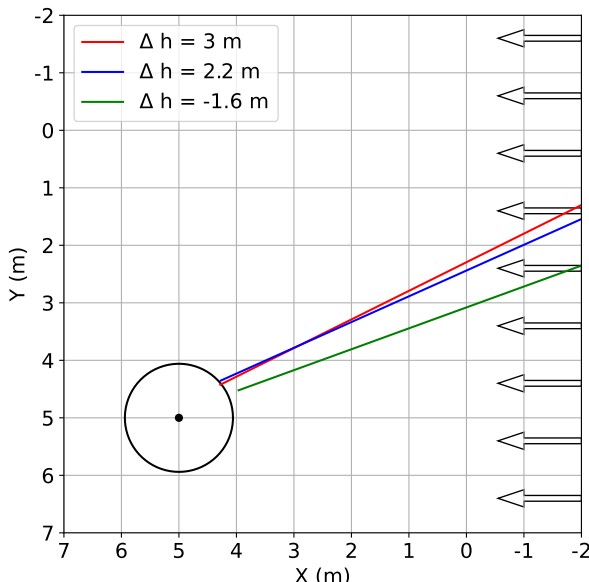

**Figure 10.** A top-down view of line scans after the coordinate has been centered at the average drone center (black dot) with the drone about $3\,\text{m}\,(4.2\,D)$ and $2.2\,\text{m}\,(3.1\,D)$ below and $1.6\,\text{m}\,(2.3\,D)$ above the line. The black circle represents the drone's diameter of $1.88\,\text{m}$.

that is most relevant for placing anemometers, both for plane and line scans. Consequently, CFD simulations can be a reliable, convenient, and affordable approach to studying complex flow around drones under different flow conditions.

Based on the aforementioned findings, we display the flow patterns on a vertical plane by CFD simulations with the lowest observed free-stream velocity of $1.3\,\text{m}\,\text{s}^{-1}$ in Fig. 13. The small free-stream velocity chosen represents the most critical sce-
270   nario under which the significant impact from PIF can be expected for sensor placement in the forward direction (Wen et al., 2019). With stronger wind speeds, the drone's downwash becomes more tilted. When a sonic anemometer is placed $0\,\text{m}$ below the mean rotor plane, the horizontal distance should be $4.8\,\text{m}\,(6.8\,D)$ upstream to achieve less than $\pm5\%$ distortion of both horizontal and vertical wind velocities at lower ambient speed. For many wind energy applications, however, the system will operate at ambient wind speeds above $4\,\text{m}\,\text{s}^{-1}$, reducing the required distance from the drone considerably.

275  **6  Discussions and conclusions**

Based on the CFD simulations presented by Ghirardelli et al. (2023), we primarily evaluated low wind conditions as the worst-case scenario for drone-mounted sensor placement in this novel study. Three ground-based continuous-wave Doppler lidars with high spatial and temporal resolution were applied to characterize the propeller-induced flow generated by a rotary-wing drone hovering at different heights. Through the synchronization of the three lidars, two scanning scenarios were designed:
280   plane scanning and line scanning. The wind fields retrieved by the lidars were compared with those obtained from CFD simulations. Both plane and line scans show a good agreement with the simulations.

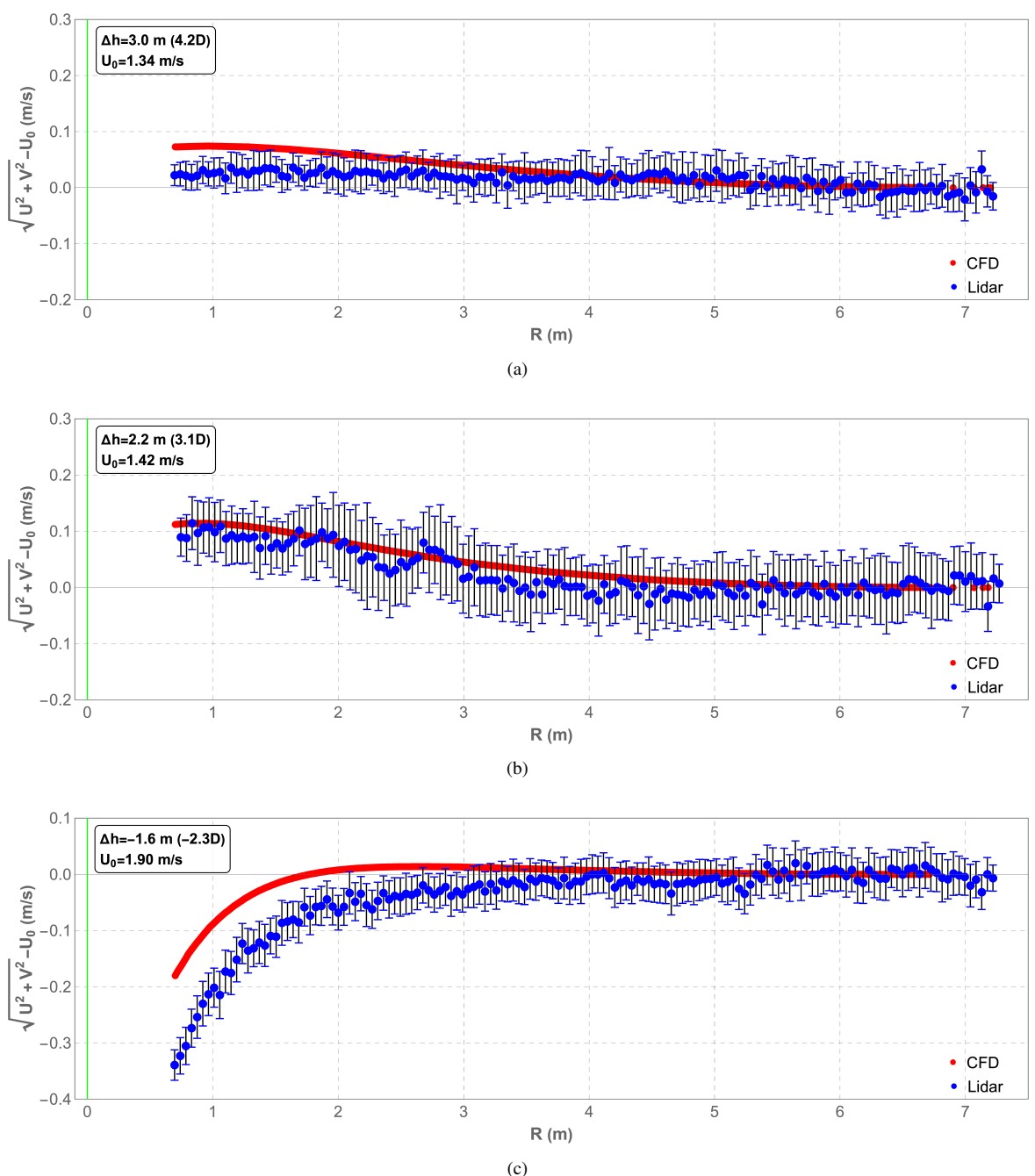

**Figure 11.** Comparison of normalized horizontal wind velocity between CFD simulations (red lines) and 3-min-average lidar data (blue dots) along the scanning line with the drone at three different heights. **(a)** The drone is about $3\,\mathrm{m}$ ($4.2\,D$) below the line. **(b)** The drone is about $2.2\,\mathrm{m}$ ($3.1\,D$) below the line. **(c)** The drone is about $1.6\,\mathrm{m}$ ($2.3\,D$) above the line. The green line indicates the average drone position and the error bars are $\pm 3$ times the standard error of the mean.

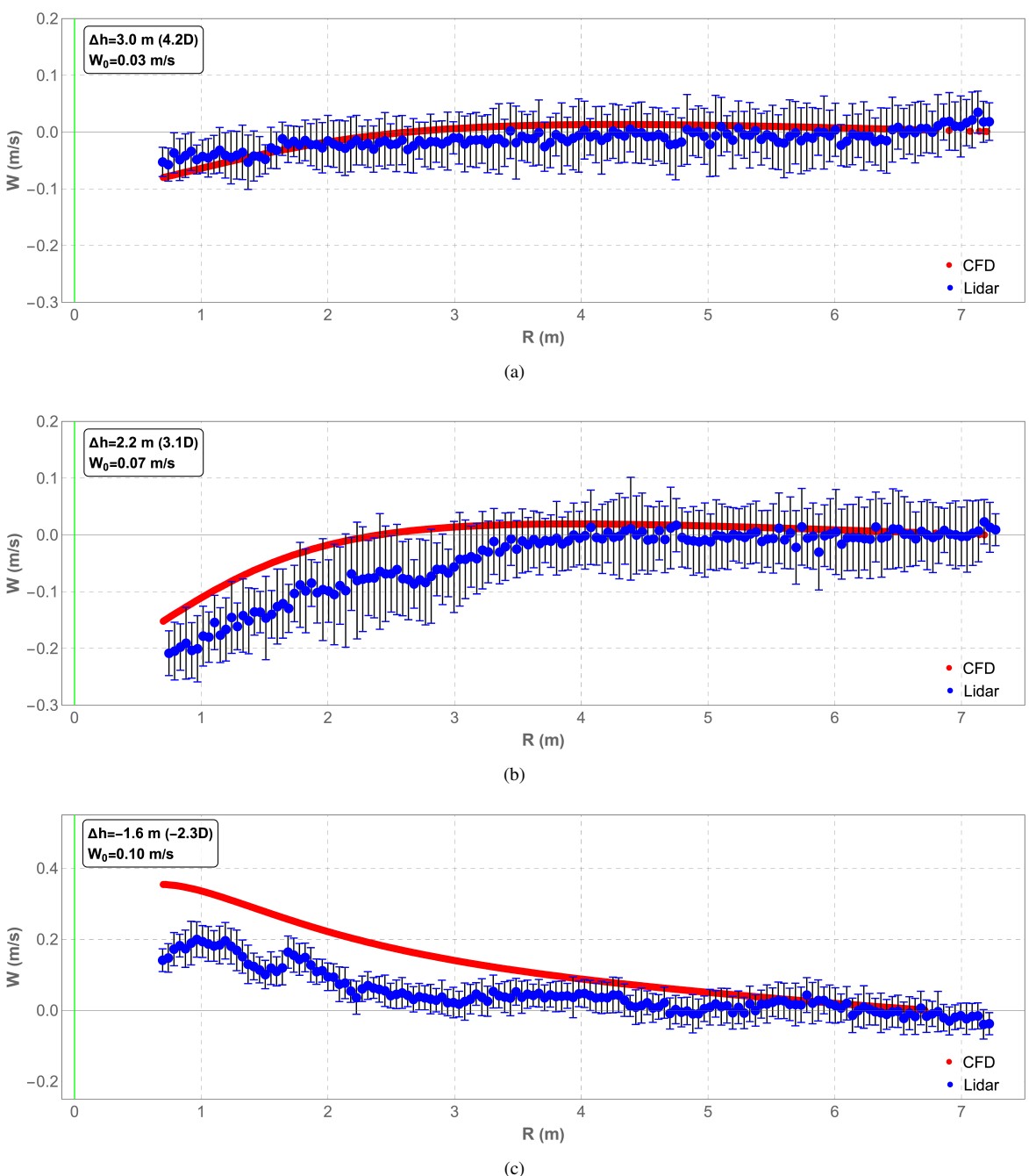

**Figure 12.** Comparison of normalized vertical wind velocity between CFD simulations (red lines) and 3-min-average lidar data (blue dots) along the scanning line with the drone at three different. **(a)** The drone is about $3\,\text{m}$ ($4.2\,D$) below the line. **(b)** The drone is about $2.2\,\text{m}$ ($3.1\,D$) below the line. **(c)** The drone is about $1.6\,\text{m}$ ($2.3\,D$) above the line. The green line indicates the average drone position and the error bars are $\pm 3$ times the standard error of the mean.

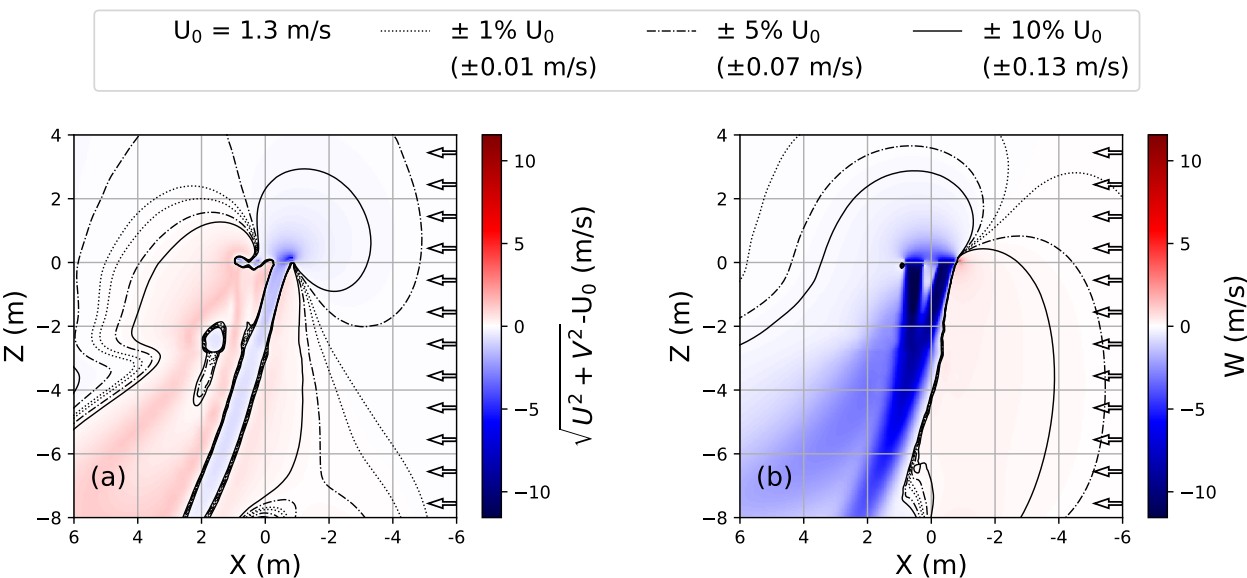

**Figure 13.** Flow patterns and velocity contours of $1\%$, $5\%$, and $10\%$ deviation from $U_0 = 1.3\,\mathrm{m\,s^{-1}}$ on cross-sections over the $xz$-plane by CFD simulations. **(a)** Normalized horizontal wind velocity. **(b)** Vertical wind velocity.

For the plane scans, lidar measurements and CFD simulations show similar flow patterns. In the case of the plane scan at $0.7\,\mathrm{m}$ ($1\,D$) below the drone, both methods demonstrate that the disturbance zone ($1\%$ difference relative to the free-stream wind velocity) for the horizontal wind velocity stretches $2\,\mathrm{m}$ ($2.8\,D$) from the drone center and more than $5\,\mathrm{m}$ ($7D$) for the vertical component. However, the CFD simulations show larger drone-induced peak velocity deviations from the free flow than the lidar does, which can be explained by considering the unresolved atmospheric turbulence, the principle of lidar measurements, and the averaging effect by the drone's random drift. Additionally, the line scans show that the velocity difference between the two methods is about $0.1\,\mathrm{m\,s^{-1}}$ (less than $4\%$ relative to the free-stream velocity) at low ambient wind speeds.

It is worth noting that there is still a disturbance in the vertical wind component of about $1\%$ even at an upwind distance of $5\,\mathrm{m}$, with the drone's diameter of $1.88\,\mathrm{m}$ and a propeller size of $0.71\,\mathrm{m}$. Mounting a 5-meter boom on such a drone is virtually impractical. However, this considerable distance is the result of selecting a stringent threshold of $1\%$ disturbance. For wind speeds lower than $4\,\mathrm{m\,s^{-1}}$, $1\%$ velocity deviation is less than many full-size sonic anemometers' nominal sensitivity. With a less strict threshold of $5\%$ velocity deviation for both horizontal and vertical winds, this 5-meter distance can be substantially brought down to 2 meters when a background flow of at least $4\,\mathrm{m\,s^{-1}}$ is present, corresponding to a flow distortion of $\pm 0.2\,\mathrm{m\,s^{-1}}$ (Fig. 8e). This deviation is similar to the accuracy reported by Wetz et al. (2021) as well as Wildmann and Wetz (2022).

As drone-based sonic anemometer applications are still in the early stages of development, our study serves as a proof-of-concept for broader and more complex future research. It will be necessary to study various wind conditions and the impact of drone-mounted wind sensors on turbulence measurements. Furthermore, a full-size sonic anemometer could be mounted on

the drone in the upstream direction to validate the potentially optimal position defined by CFD simulations and to benchmark wind velocity uncertainties with an external sonic anemometer.

*Data availability.* Data underlying the results presented in this paper can be obtained from the authors upon reasonable request.

*Author contributions.* All authors have contributed to the paper preparation. Conceptualization, JM, JR, MS; methodology, project management, and experiment conduction, LJ, MG, JM, MS, JR, SK; data analysis, LJ, MG; writing—original draft preparation, LJ, MG; writing—review and editing, LJ, MG, JM, MS, JR, SK. All authors have read and agreed to the published version of the manuscript.

*Competing interests.* The authors declare no conflict of interest.

*Acknowledgements.* The authors would like to thank Michael Courtney, Per Hansen, Lars Christensen, Michael Rasmussen, and Claus Brian Munk Pedersen from DTU, for their helpful support during the field experiment, fruitful discussions, and helpful comments.

*Financial support.* This research is mainly funded by the two Marie Sklodowska Curie Innovation Training Networks LIKE (H2020-MSCA-ITN-2019, Grant number 858358), and Train2Wind (H2020-MSCA-ITN-2019, Grant number 861291). Additional funding comes from Atmospheric FLow, Loads, and pOwer for Wind energy (FLOW, HORIZON-CL5-2021-D3-03-04, Grant number 101084205), funded by the European Union.

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
