# Peer review of "Rotary-wing drone-induced flow – comparison of simulations with lidar measurements"

_EGUsphere, 2023_

## Author Comment (AC1)

Responses to Reviewers' Comments for Manuscript

**Rotary-wing drone-induced flow – comparison of simulations with lidar measurements.**

Addressed Comments for Publication to

Atmospheric Measurement Techniques

by

Liqin Jin, Mauro Ghirardelli, Jakob Mann, Mikael Sjöholm, Stephan T. Kral, and Joachim Reuder

Dear Reviewers,

Please find enclosed the revised version of our manuscript entitled "Rotary-wing drone-induced flow – comparison of simulations with lidar measurements". We would like to express our gratitude to you both for providing valuable comments that have greatly contributed to the improvement of our work. In this revision, we have carefully considered and addressed the comments raised, aiming to enhance the quality and clarity of the manuscript.

Below, we provide a summary of the main modifications made and a detailed point-by-point response to your comments. We sincerely appreciate your time and effort in reviewing our manuscript, and we are grateful for the opportunity to address your concerns.

Sincerely,

Liqin Jin, Mauro Ghirardelli, Jakob Mann, Mikael Sjöholm, Stephan T. Kral, and Joachim Reuder

**Authors' Response to Reviewer I**

> **General Comments.** The manuscript discusses the use of ultrasonic anemometers mounted on rotary-wing drones as a potentially cost-effective alternative to traditional meteorological mast-mounted anemometers for wind energy applications. However, concerns are raised about the accuracy of wind velocity measurements due to propeller-induced flow disturbances. The study presents an experiment using three short-range continuous-wave Doppler lidars (DTU WindScanners) to measure the complex and turbulent three-dimensional wind field around a hovering drone at low ambient wind speeds. The results from lidar measurements are compared to computational fluid dynamics (CFD) simulations to validate the accuracy of drone-mounted wind sensors. While the data and measurements are promising, the manuscript has several weaknesses.

**Response:**

Dear Reviewer, we are glad that you find our research valuable and the results are promising. After reading through your comments, we implemented changes to our manuscript including your suggestions. We will now address, one by one, your comments and suggestions.

> **Comment 1**
>
> - Clarify the novelty of this research in comparison to existing studies using drones with anemometers.
>
> - Specify whether the proposed method complements or extends current techniques in wind field measurements.
>
> - Enhance the logical relationships between references, discussing the limitations of previous research and their relevance to the current study.
>
> - Clearly state the contributions of this study regarding the development of advantages for rotary-wing drones in wind field measurements.

**Response:**

Thank you for pointing these out. We revised our paper to address this concern. We decided to combine these four points because we found a common theme: the start of the study was unclear. For this reason, we have critically revised the introduction to explain how our research advances the current state of knowledge in the use of drones with anemometers for atmospheric turbulence characterization, and also in the wind measurement field.

Our study is distinct in several key ways:

- Comprehensive approach: Our research aims at the study of full-size sonic anemometers mounted on rotary-wing UAVs, while previous studies have often focused on using lightweight anemometers or have primarily measured mean horizontal wind or vertical ABL wind profiles. This can potentially lead to more accurate measurements of three-dimensional turbulent wind velocity, which is crucial for a detailed understanding of atmospheric dynamics.

- Experimental validation: we provide experimental validation of the CFD simulation model proposed by Ghirardelli et al. [2023]. This CFD model can be further used to optimize drone-mounted sensor placement to minimize the influence of propeller-induced flow (PIF). The validation approach presented in this manuscript

helps bridge the gap between theory and real-world application, offering robust data and insights into the characteristics of PIF around large multi-copter drones.

- Innovative Application and Mounting Strategy for Standard Sonic Anemometers: While our study utilizes commercially standard sonic anemometers, we introduce a novel application by configuring these sensors in an upwind-facing, boom-mounted arrangement beneath the drone's fuselage. This configuration is unique in the context of drone-based atmospheric measurements.

- Use of CW Doppler lidars: Beyond simulations, we employ high-resolution continuous-wave (CW) Doppler lidar to investigate PIF experimentally. CW lidars can provide accurate, three-dimensional flow observations, to validate CFD simulations that will be further used to refine sensor placement for optimal data collection by drone-mounted sensors. To the best of our knowledge, this is the first study utilizing three synchronized CW Doppler lidars to investigate the turbulent three-dimensional flow around a rotary-wing drone and compare with CFD simulations.

We believe these elements collectively represent a substantial advancement in the field of atmospheric measurements using UAVs and offer valuable insights and methodologies for future research. We hope this clarification underscores the innovative aspects of our work and its contribution to the broader scientific community.

> **Comment 2**
>
> Address the variations in flow disturbances and spatial flow fields induced by different types of rotary-wing drones.

**Response:**

Thank you for your valuable feedback and for highlighting the importance of addressing variations in flow disturbances and spatial flow fields induced by different types of rotary-wing drones. We appreciate the opportunity to discuss this further. We fully acknowledge the significance of understanding these variations to provide a comprehensive overview of the propeller-induced flow (PIF) dynamics. However, we believe there is a notable

lack of standardized methods and sufficient data for comparative analysis across various drone models and this is still in progress. Our approach, focusing on a single case (eight rotors in a contra-rotating set-up), was chosen to provide an in-depth understanding of propeller-induced flow for this specific drone type that is suited for lifting heavy payloads. We believe that establishing a detailed baseline for one drone can serve as a valuable reference point for future comparative studies. We therefore aim to contribute a foundational piece to the broader puzzle on which future research can be built.
* * *
**Comment 3**

Consider using more conventional symbols to represent horizontal wind speed components to improve readability and understanding.
* * *
**Response:**

Thank you very much for pointing this out. We agree with this comment and have used the conventional symbols $U_h$ to represent horizontal wind speed components. This is added in **L220** in the revised manuscript as For a clearer illustration, we normalized the horizontal wind velocity $U_h = \sqrt{U^2 + V^2}$ by subtracting the free-stream wind speed $U_0$.
* * *
**Comment 4**

Provide clearer explanations for critical aspects such as sampling frequency, sampling time, and numerical simulation parameters.
* * *
**Response:**

Thank you very much for this point. We have explained more clearly about sampling frequency, and sampling time in **L145**. The new sentence is After a block averaging of 726 spectra to reduce noise fluctuations, the final spectrum is sampled at a frequency of $322 \text{ Hz} = (120 \text{ MHz})/(512 \cdot 726)$ with the corresponding sample time of 3.1 milliseconds for each spectrum. For numerical simulation parameters, we used default setups, which is indicated in **L124** as Following the initial mesh setup and selection of the turbulence

model, we used standard settings from Ansys Fluent to ensure consistency and reliability.

> ### Comment 5
> Address the potential reliability issues of averaging radar wind measurements in terms of error analysis and experimental design.

**Response:**

Thank you very much for this point. We agree and have addressed this comment in **L133**, which is The use of CW Doppler lidars is beneficial for a variety of wind energy applications. In spite of this, CW Doppler lidars are susceptible to moving objects away from the intended focus point, such as flying birds. Besides, their spatial resolution decreases as the focus distance increases, which may deteriorate the accuracy of wind velocity and turbulence measurements by CW lidars [Jin et al., 2022]. Therefore, we placed the three lidars as close to the intended scanning positions as possible to compact the measurement volume [Angelou et al., 2012] and minimize potential biases resulting from volume-averaging [Clive, 2008, Sjöholm et al., 2009, Forsting et al., 2017]. To improve the accuracy of flow velocity retrieval, we discard spectra containing Doppler shifts caused by hard targets and out-of-focus moving objects during the post-processing.

> ### Comment 6
> Explain the rationale behind choosing three radar wind devices and their arrangement, considering potential sources of error.

**Response:**

Thank you very much for this comment. We have explained the reason why we used three lidars and their arrangement in **L130** that A single CW Doppler lidar can only measure the one-dimensional projection $v_{LOS}$ of wind velocity vector along its line-of-sight beam

direction. Therefore, by combining the independent and simultaneous measurements of $v_{LOS}$ from three Doppler lidars, a full three-dimensional wind vector can be retrieved. In **L133** we explained the potential sources of error by using CW lidars as The use of CW Doppler lidars is beneficial for a variety of wind energy applications. In spite of this, CW Doppler lidars are susceptible to moving objects away from the intended focus point, such as flying birds. Besides, their spatial resolution decreases as the focus distance increases, which may deteriorate the accuracy of wind velocity and turbulence measurements by CW lidars [Jin et al., 2022]. Therefore, we placed the three lidars as close to the intended scanning positions as possible to compact the measurement volume [Angelou et al., 2012] and minimize potential biases resulting from volume-averaging [Clive, 2008, Sjöholm et al., 2009, Forsting et al., 2017]. To improve the accuracy of flow velocity retrieval, we discard spectra containing Doppler shifts caused by hard targets and out-of-focus moving objects during the post-processing.

> ### Comment 7
>
> Discuss the impact of drone-mounted wind sensors on the measurement of turbulence characteristics, in addition to average wind speed.

**Response:**

Thank you for your valuable comment. We acknowledge the importance of understanding the impact of drone-mounted wind sensors on turbulence characteristics research.

Our research is structured into three distinct phases. The first involves CFD simulations on the PIF (Ghirardelli et al. [2023]). The second phase, which was conducted in December 2022, focused on validating these CFD simulations by comparing the wind velocities obtained with those measured by lidar. The results of this phase are the subject of the current paper. The third and final phase entailed a field measurement campaign where we combined a full-size sonic anemometer with the drone to compare both wind velocity and turbulence against data from mast-mounted sonic sensors. This phase was completed

in late December 2023, and a draft detailing its findings is now in preparation.

In the revised manuscript, we calculated the velocity uncertainty based on 3-min lidar data, which is demonstrated by the error bars in Figure 11 and 12. Furthermore, we found that the flow is less turbulent as it approaches the drone since the error bars become shorter, which indicates that drone-mounted sonic sensors may impact the measurement of turbulence characteristics. However, this needs further investigations and is beyond the scope of the present study. We have implemented this comment in **L254** that The flow appears to be less turbulent as it approaches the drone, which is indicated by the shorter error bars in Fig. 11 and Fig. 12. This suggests that drone-mounted sonic sensors may impact the measurement of turbulence characteristics. However, this needs further investigations and is beyond the scope of the present study. as well as in **L290** that It will be necessary to study various wind conditions as well as the impacts of drone-mounted wind sensors on turbulence measurements.

> #### Comment 8
>
> Improve English language expression, particularly regarding sentence structure and readability.

**Response:**

Thank you very much to point this out. We agree with this comment and have carefully checked the whole manuscript to improve English language expression. Hope this time it can be accepted by you.

**References**

N. Angelou, J. Mann, M. Sjöholm, and M. Courtney. Direct measurement of the spectral transfer function of a laser based anemometer. *Review of scientific instruments*, 83(3): 033111, 2012.

P. Clive. Compensation of vector and volume averaging bias in lidar wind speed measurements. In *IOP Conference Series: Earth and Environmental Science*, volume 1(1), page 012036. IOP Publishing, 2008.

A. M. Forsting, N. Troldborg, and A. Borraccino. Modelling lidar volume-averaging and its significance to wind turbine wake measurements. In *Journal of Physics: Conference Series*, volume 854(1), page 012014. IOP Publishing, 2017.

M. Ghirardelli, S. T. Kral, N. C. Müller, R. Hann, E. Cheynet, and J. Reuder. Flow structure around a multicopter drone: A computational fluid dynamics analysis for sensor placement considerations. *Drones*, 7(7):467, 2023.

L. Jin, J. Mann, and M. Sjöholm. Investigating suppression of cloud return with a novel optical configuration of a doppler lidar. *Remote Sensing*, 14(15):3576, 2022.

M. Sjöholm, T. Mikkelsen, J. Mann, K. Enevoldsen, and M. Courtney. Spatial averaging-effects on turbulence measured by a continuous-wave coherent lidar. *Meteorologische Zeitschrift (Berlin)*, 18, 2009.

---

## Author Comment (AC2)

Responses to Reviewers' Comments for Manuscript

**Rotary-wing drone-induced flow – comparison of simulations with lidar measurements.**

Addressed Comments for Publication to

Atmospheric Measurement Techniques

by

Liqin Jin, Mauro Ghirardelli, Jakob Mann, Mikael Sjöholm, Stephan T. Kral, and Joachim Reuder

Dear Reviewers,

Please find enclosed the revised version of our manuscript entitled "Rotary-wing drone-induced flow – comparison of simulations with lidar measurements". We would like to express our gratitude to you both for providing valuable comments that have greatly contributed to the improvement of our work. In this revision, we have carefully considered and addressed the comments raised, aiming to enhance the quality and clarity of the manuscript.

Below, we provide a summary of the main modifications made and a detailed point-by-point response to your comments. We sincerely appreciate your time and effort in reviewing our manuscript, and we are grateful for the opportunity to address your concerns.

Sincerely,

Liqin Jin, Mauro Ghirardelli, Jakob Mann, Mikael Sjöholm, Stephan T. Kral, and Joachim Reuder

**Authors' Response to Reviewer II**

> **General Comments.** Jin et al. present a method to determine the flow around a rotary-wing UAS in the field. Such experiments are extremely valuable, and the solution to use a triple-Doppler short-range lidar setup is innovative and unique. It is very difficult to get such data in other ways. Despite the very good concept, I think the study misses a lot of opportunities for a more profound analysis. From the manuscript I cannot conclude if CFD simulations are good enough to study the flow around a multicopter well enough for sensor placement in all relevant conditions. It also remains unclear if the results can be transferred to other conditions and other platforms. I can only recommend the manuscript for publication in AMT after major revision. I give some general and specific comments below:

**Response:**

Dear Reviewer, we appreciate the time and effort that you have dedicated to providing your insightful comments on our paper. After reading through your comments, we implemented our manuscript including your suggestions. We will now address, one by one, your comments and suggestions.

> **Comment 1**
>
> The authors motivate the work with wind energy research. The experiment is carried out at wind speeds below 4.1 m/s. Values of turbulence intensity are not even given. These are not conditions that are relevant for wind energy. I understand that these are conditions that are critical for sensor placement, but if the application is at higher wind speeds, it would be important to know if the sensor placement can be changed if low wind speeds are disregarded.

**Response:**

Thank you for your insightful comments. For wind energy applications, high wind speeds are essential, and our previous study concluded that high wind speeds facilitate optimal

sensor displacement upwind, as the downwash is pushed downstream. However, we believe that understanding sensor behavior at low wind speeds is equally important, especially to ensure accurate measurements across a variety of meteorological conditions. Drone-induced flow is a distinct challenge associated with low wind speeds, making them a worse scenario that must be studied in order to develop comprehensive sensor placement strategies.

In addition, while wind energy is undoubtedly a key domain for the application of our drone-based research, our overarching aim is broader. We seek to develop a multicopter-based system capable of resolving turbulence across a wide spectrum of wind speeds and atmospheric conditions. This was not stated clearly in the previous version of the manuscript. Therefore, we have expanded the scope of our paper to include a comprehensive study of boundary layer meteorology.

We have implemented this comment in the manuscript now by modifying the introduction part. We have revised the first paragraph as The proper characterization of atmospheric turbulence is essential to understanding the structure and dynamics of the atmospheric boundary layer (ABL) [Stull, 1988, Wyngaard, 2010]. It is therefore crucial to obtain accurate wind and temperature measurements with high spatial and temporal resolution for a variety of basic and applied ABL research topics, such as weather and climate prediction [Teixeira et al., 2008], wind energy meteorology [Emeis, 2010, Albornoz et al., 2022], and atmospheric modelling [Etling, 1996].

> ### Comment 2
> To my understanding all the analysis is based on two short measurement periods. This seems very little in order to draw general conclusions.

**Response:**

Thank you for pointing this out. We acknowledge that our study is based on two relatively short measurement periods. As we mentioned in the manuscript the drone has a nominal maximum flight time of 45 minutes. As the field experiment was conducted at low

temperatures, the maximum flight time was reduced considerably. In order to maintain the same ambient conditions and calibration of the drone system, we decided to measure the flow at different heights during each of the flights, instead of doing one position for each flight. This ended in us positioning the drone at 4 different heights for 5 minutes each. Then, we analyzed all the lidar data we had during the post-processing of plan scans, involving three- to four-minute data. For the line scans, we analyzed 3-min lidar data instead of 1-min data. From the error bars in Figure 11 and 12, we found the results by CFD simulations and lidar measurements are comparable.

We understand that more extended measurement periods may provide a more robust dataset and allow for more definitive conclusions. It is also important to consider the fluctuations in wind speed, which may lead to bias if the measurements are taken over a long period of time.

**Comment 3**

One main difference between the field experiment and the CFD which it is compared to is in my opinion the turbulence. The authors say that the k-epsilon model is used in RANS, but do not give details about the parameters that are set in the model setup (i.e. boundary conditions for k and epsilon) and how they compare to the measurements (I did not find it in Ghirardelli 2023 either). How do the results depend on the turbulence settings?

**Response:**

Thank you very much for this comment. In our CFD analysis, we employed the standard k-$\epsilon$ turbulence model in ANSYS Fluent, utilizing its default parameter settings. We made this decision based on some important considerations that are both crucial to the integrity and relevance of our research. Firstly, the use of standardized settings offers the advantage of consistency, fostering comparability with other studies employing similar methodologies. As the default parameters provide a pragmatic balance between customization and realism, they are a pragmatic solution to the complexity of adjusting

k and $\epsilon$ values for each unique simulation scenario. Therefore, we added this point in the manuscript in **L124** as Following the initial mesh setup and selection of the turbulence model, we used standard settings from Ansys Fluent to ensure consistency and reliability.

Additionally, our study goes beyond the turbulence modeling to encompass more overarching flow characteristics and systemic behaviors. We have calculated the turbulence intensity (TI) based on the lidar data, which is 0.04. This means that, in achieving our research objectives, complex adjustments to turbulence model parameters are unlikely to yield significant improvements.

> **Comment 4**
>
> I think an opportunity is missed to measure the downdraft with the sonic anemometers and the short-range lidars at the same time with a lower hover altitude above the sonics. This would give more confidence in the measurements and the comparison to the simulations.

**Response:**

Thank you for this nice comment. We agree. However, the sonic anemometer on the small met mast used in this study doesn't work. In our previous study [Jin et al., 2023], we focused three CW lidars very close to a mast-mounted sonic anemometer and one lidar's probe length is about 1.2 m, which is slightly larger than that of this study (0.56 to 0.87 m). The 50 Hz times series of wind velocity measured by lidars and the sonic anemometer agree very well in Figure 11, Table 4 and 5. The velocity differences relative to the sonic velocity could be 0.18% at no precipitation period. Therefore, as validation method, we used three lidars to compare wind velocity on horizontal planes and lines with CFD simulations, which is not possible with single-point measurements from sonic anemometers.

> ## Comment 5
> Is it really necessary to describe how wind is retrieved from Doppler lidar spectra in this study?

**Response:**

Thank you very much for pointing this out. We agree with this comment and delete the wind velocity retrieval process in the manuscript.

> ## Comment 6
> Section 2.2 I think that the WindScanner system is well described in the literature and since it is only used for validation measurements here, a specification of the uncertainty and reference to the literature would be sufficient. The data processing description could be omitted in my opinion.

**Response:**

Thank you very much for this comment. We agree with this point and deleted the detailed introduction of WindScanner system. Instead, we explained the reason why we use three lidars and the potential measurements errors. The new paragraph from **L128** to **L136** is The ground-based, short-range WindScanner system developed by DTU Wind and Energy Systems consists of three synchronized coherent CW Doppler lidars (Fig. 3), which are capable of accurately retrieving wind vectors and measuring turbulence [Sjöholm et al., 2009, Mikkelsen et al., 2020, Jin et al., 2023]. A single CW Doppler lidar can only measure the one-dimensional projection $v_{LOS}$ of wind velocity vector along its line-of-sight beam direction. Therefore, by combining the independent and simultaneous measurements of $v_{LOS}$ from three Doppler lidars, a full three-dimensional wind vector can be retrieved.

The use of CW Doppler lidars is beneficial for a variety of wind energy applications. In spite of this, CW Doppler lidars are susceptible to moving objects away from the intended focus point, such as flying birds. Besides, their spatial resolution decreases

as the focus distance increases, which may deteriorate the accuracy of wind velocity measurements by CW lidars [Jin et al., 2022b]. Therefore, we placed the three lidars as close to the intended scanning positions as possible to compact the measurement volume [Angelou et al., 2012] minimize potential biases resulting from volume-averaging [Clive, 2008, Sjöholm et al., 2009, Forsting et al., 2017]. To improve the accuracy of flow velocity retrieval, we discard spectra containing Doppler shifts caused by hard targets and out-of-focus moving objects during the post-processing [Jin et al., 2022a].

Because the other reviewer asked us to provide clearer explanations for critical aspects such as sampling frequency, sampling time, and numerical simulation parameters. Therefore, we would like to keep he data processing description. Hope this can be accepted by you.
* * *
**Comment 7**

Is a 5-m distance to the UAV a realistic position for a sensor in flight? What does that mean for the weight of the system, the flight time, the stability?
* * *
**Response:**

Thank you for your insightful comment. We acknowledge that positioning a sensor 5 meters away from the UAV is not practical for in-flight operations due to significant implications on the system's weight, flight time, and stability. This recommendation stems from CFD simulations and lidar measurements under very low wind speeds conditions, aiming at achieving less than 1% velocity difference relative to the free flow, both horizontally and vertically. As wind speed increases, the aerodynamic interference from the drone diminishes, allowing for closer sensor placement without compromising the accuracy significantly. Therefore, in more typical conditions with higher wind speeds, the required distance for accurate measurements would be considerably less.

It is also important to mention that the 1% distortion threshold is particularly stringent, especially when considering the inherent measurement inaccuracies of state-of-the-art sensors in case of low wind. This level of precision, although ideal, is beyond the capabilities of current technology in most practical scenarios. Thus, while our study

provides a theoretical basis for optimal sensor placement under various conditions, the real-world applicability must consider a balance between ideal accuracy and operational feasibility.

We have implemented this comment in the revised manuscript in **L283**, which is It is worth noting that there is still a disturbance in the vertical wind component of about 1% even at an upwind distance of 5 m, with the drone's diameter of 1.88 m and a propeller size of 0.71 m. Mounting a 5-meter boom on such a drone is virtually impractical. However, this considerable distance is the result of selecting a stringent threshold of 1% disturbance. For wind speeds lower than 4 ms$^{-1}$, 1% velocity deviation is less than many full-size sonic anemometers' nominal sensitivity. Therefore, by using a less strict threshold, this 5-meter distance can be substantially reduced to 2 m, which is also possible at high wind speeds due to the propeller-induced flow shifting downstream.

> ### Comment 8
> p.1, l.6: maybe introduce the CFD simulations first, before comparing to measurements.

**Response:**

Thank you very much for pointing this out. We agree with this point and have introduced CFD simulations first in **L4**. The new sentence is However, propeller-induced flow may deteriorate the accuracy of free wind velocity measurements by wind sensors mounted on drones, which needs to be investigated for optimal sensor placement. Computational fluid dynamics (CFD) simulations are an alternative to experiments for studying characteristics of the propeller-induced flow, but require validation. Besides, we also changed the structure in Section 2 by introducing CFD simulations setup first.

> ### Comment 9
> p.2, l.41: what is a reference for the extensive CFD studies?

**Response:**

We appreciate that you point this out. We have unintentionally used the wrong phrase here. We acknowledge that CFD is a prevalent tool in UAV research. However, the specific exploration of PIF, particularly in the context of sensor placement, is not as extensively studied. We have revised the introduction section of our manuscript to more accurately reflect the current state of research in this area. This amendment clarifies that our study addresses a unique aspect of UAV research, specifically focusing on optimizing sensor placement in relation to PIF, while majority of studies have primarily examined how PIF impacts drones' flight stability for design purposes [Zheng et al., 2018, Guillermo et al., 2018, Lei and Lin, 2019, Guo et al., 2020].

We believe this revision provides a more accurate representation of the literature and better contextualizes the contribution of our work by filling a research gap in this field. We are grateful for your insightful comments, which have undoubtedly helped improve the quality and accuracy of our manuscript.

> **Comment 10**
>
> p.2, l.45f: I think it would be better to give a distance relative to the rotor diameter, like in the text before.

**Response:**

Thank you very much for this comment. We agree with this point and add the information of the distance relative to the rotor diameter. Now the new sentence in **L50** is Vasiljević et al. [2020] presented a proof-of-concept drone–lidar system and concluded that the lidar should be placed out of the drone's disturbance zone stretching between 1 and 2 m (1.9 $D$ and 3.7 $D$ with rotor diameters of 0.53 m) from the center, based on measurements of radial wind speed.

> **Comment 11**
>
> p.6, l.130f: It would be nice to know upfront what is the purpose of plane and line scan.

**Response:**

Thank you very much to point out this comment. We agree and have added one sentence to explain the reason to have plane and line scans in **L160**, which is The flow-field evolution in a horizontal plane at a certain distance below the drone was measured for comparison with the CFD simulations, while fast line scans enable detailed comparisons in the upstream direction, which is the most promising region for a boom-mounted sonic anemometer attached to the drone Ghirardelli et al. [2023].

> ### Comment 12
>
> p.7, ll.134ff: I do not quite understand the reasoning for that pattern. If the lidars had just scanned a simple square with 20 lines (corresponding to the 20 orange dot lines) successively in y-direction, the scan would only last 20 s in my understanding. Would that not be better?

**Response:**

Thank you very much to point this out. It would have been good to have a shorter scanning cycle. However, if design a square with some lines successively in y-direction, we have to rotate the dual prisms very fast, which is explained in **L168** as The three laser beams followed a path that started from one side of the line, passed through the disc center to the other end of the line, and then returned over the same route to its starting point. This allows the dual-prism lidars to complete the pattern in a reasonably short time, if the symmetry axis of each lidar crosses the disc center and the trajectory in the vicinity passes exactly through the same crossing point.

> ### Comment 13
>
> p.7, l.142: Was RTK GPS used in that case as mentioned in the drone description? I would not expect the drift in that case.

**Response:**

Yes, RTK GPS was indeed utilized as part of our drone's setup, as mentioned in the

description. We appreciate your observation regarding the unexpected drift despite using RTK GPS. It appears that there might be some issues within our setup that led to this drift. The positioning of the drone can of course be improved by a better position estimate but it depends to a large degree on the tuning of the drone, which has admittedly not been optimal for this purpose. Ensuring accurate and reliable data collection is our top priority, and as part of this, we are committed to improving the drone's tuning.

> ### Comment 14
>
> p.10, l.176 and Fig. 7: what is the "certain threshold" and how is it determined. The numbers in Fig.7c are much higher than in 7d. While it seems obvious for 7c that hard target hits are possible, why are the areas in 7d so broad?

**Response:**

Thank you for this comment. The "certain threshold" is determined based on the sum of 1-min averaged power spectral density. From Figure 1(a), we can clear see that when the lidars hit the drone, the sum of the spectrum is much higher and the relatively lower values are obtained from aerosol-induced spectra. Therefore, we apply the threshold as 2 to remove spectra with very high peaks and the result is shown in Figure 1(b).

[Figure]

Figure 1: Sum of 1-min-average spectrum of three lidars at height $\Delta h = -0.7$ m. **(a)** Before applying threshold to remove spectra with very high peaks. **(b)** After applying threshold to remove spectra with very high peaks. The yellow indicates the threshold value, which is 2 in this case.

**The numbers in Fig.7c are much higher than in 7d is because the drone in Fig.7d is much more out of focus than that in Fig.7c.** Considering the probe length of the three lidars ranges from 0.56 to 0.87 m, the laser intensity drops sharply from the scanning plane and the drone in Fig.7d is further away from the focus point. The Doppler spectrum with the highest peak value and the sum of all the spectra on the scanned plane measured by lidar 3 at height $\Delta h = -0.7$ m and $\Delta h = -2.1$ m are shown in Figure 2. For Figure 2(a) where the drone is only 0.7 m above the scanning plane, the maximum spectral power is about 3000, while for Figure 2(b) where the drone is 2.1 m above the scanning plane, the highest value is about 30.

[Figure]

Figure 2: Spectrum with the highest peak value and the sum of all spectra on the scanned plane measured by lidar 3 at two heights. **(a)** Spectrum at height $\Delta h = -0.7$ m. **(b)** Spectrum at height $\Delta h = -2.1$ m. **(c)** Sum of all spectra at height $\Delta h = -0.7$ m. **(d)** Sum of all spectra at height $\Delta h = -2.1$ m.

**Comment 15**

p.10, l.188: Can you also give information about turbulence intensity?

**Response:**

Thank you for this comment. We understand the importance of this parameter in comprehensively analyzing wind patterns and their impact on our study. In response to your request, we have calculated the turbulence intensity for the plane scan and have included it in our report. You can find the detailed information in the revised manuscript

**L217**. The updated text reads: The free-stream wind velocity was 4.09 ms$^{-1}$ along the positive $x$-axis with a turbulence intensity (TI) of 0.04 and the drone was yawed about 21.2° to the incoming wind.
* * *
**Comment 16**

p.12, l.206: can the differences in the comparison between CFD and measurement also be due to effects of unmodelled atmospheric turbulence?

**Response:**

Thank you for this comment. Yes, we agree with this point and have implemented it in the manuscript now. The new sentences in **L224** and **L234** are It should be noted, however, that lidar measurements did not capture the very detailed flow feature due to the effects of unresolved atmospheric turbulence, the averaging effect of lidars' measurement volume, and the drone drifting approximately ±0.5 m along the free flow direction from its average position (the central black dot in Fig. 8). and This is probably due to the aforementioned effects of unsolved atmospheric turbulence, the averaging effect of lidars, and the drones' drifting.
* * *
**Comment 17**

p.15, l.227: Please also provide the distance relative to the drone size.

**Response:**

Thank you for pointing this out. We agree with this comment and incorporated this suggestion in **L264**. The revised sentence is With stronger wind speeds, the drone's downwash becomes more tilted. When a sonic anemometer is placed 0 m below the drone, the horizontal distance should be 4.8 m (6.8 $D$) upstream to achieve less than ±5% distortion of both horizontal and vertical wind velocities at lower ambient speed. We also added all the distance relative to the rotor diameter in the manuscript.

> **Comment 18**
>
> p.15, l.235: Maybe repeat what is the criteria for the disturbance zone.

**Response:**

Thank you very much for this suggestion. We agree with this comment and have added the criteria for the disturbance zone in the manuscript. The revised sentences are In the case of the plane scan at 0.7 meters (1 $D$) below the drone, both methods demonstrate that the disturbance zone (1% difference relative to the free-stream wind velocity) for the horizontal wind velocity stretches 2 meters (2.8 $D$) from the drone center and more than 5 meters (7 $D$) for the vertical velocity component in the Conclusion part and Both methods conclude that the disturbance zone (defined by a relative deviation from the mean free-stream velocity by more than 1%) on a horizontal plane 1 $D$ (rotor diameter $D$ of 0.71 m) below the drone, extends about 2.8 $D$ upstream from the drone center for the horizontal wind velocity and more than 7 $D$ for the vertical wind velocity in the abstract.

**References**

C. P. Albornoz, M. E. Soberanis, V. R. Rivera, and M. Rivero. Review of atmospheric stability estimations for wind power applications. *Renewable and Sustainable Energy Reviews*, 163:112505, 2022.

N. Angelou, J. Mann, M. Sjöholm, and M. Courtney. Direct measurement of the spectral transfer function of a laser based anemometer. *Review of scientific instruments*, 83(3): 033111, 2012.

P. Clive. Compensation of vector and volume averaging bias in lidar wind speed measurements. In *IOP Conference Series: Earth and Environmental Science*, volume 1(1), page 012036. IOP Publishing, 2008.

S. Emeis. *Surface-based remote sensing of the atmospheric boundary layer*, volume 40. Springer Science & Business Media, 2010.

D. Etling. Modelling the vertical abl structure. *Modelling Of Atmospheric Flow Fields*, 45, 1996.

A. M. Forsting, N. Troldborg, and A. Borraccino. Modelling lidar volume-averaging and its significance to wind turbine wake measurements. In *Journal of Physics: Conference Series*, volume 854(1), page 012014. IOP Publishing, 2017.

M. Ghirardelli, S. T. Kral, N. C. Müller, R. Hann, E. Cheynet, and J. Reuder. Flow structure around a multicopter drone: A computational fluid dynamics analysis for sensor placement considerations. *Drones*, 7(7):467, 2023.

P. H. Guillermo, A. V. Daniel, and G. E. Eduardo. Cfd analysis of two and four blades for multirotor unmanned aerial vehicle. In *2018 IEEE 2nd Colombian Conference on Robotics and Automation (CCRA). IEEE*, pages 1–6, 2018.

Q. Guo, Y. Zhu, Y. Tang, C. Hou, Y. He, J. Zhuang, Y. Zheng, and S. Luo. Cfd simulation and experimental verification of the spatial and temporal distributions of the downwash airflow of a quad-rotor agricultural uav in hover. *Computers and Electronics in Agriculture*, 172:105343, 2020.

L. Jin, N. Angelou, J. Mann, and G. C. Larsen. Improved wind speed estimation and rain quantification with continuous-wave wind lidar. In *Journal of Physics: Conference Series*, volume 2265(2), page 022093. IOP Publishing, 2022a.

L. Jin, J. Mann, and M. Sjöholm. Investigating suppression of cloud return with a novel optical configuration of a doppler lidar. *Remote Sensing*, 14(15):3576, 2022b.

L. Jin, J. Mann, N. Angelou, and M. Sjöholm. Suppression of precipitation bias on wind velocity from continuous-wave doppler lidars. *EGUsphere*, 2023:1–26, 2023.

Y. Lei and R. Lin. Effect of wind disturbance on the aerodynamic performance of coaxial rotors during hovering. *Measurement and Control*, 52(5-6):665–674, 2019.

T. Mikkelsen, M. Sjöholm, P. Astrup, A. Peña, G. Larsen, M. Van Dooren, and A. K. Sekar. *Lidar Scanning of Induction Zone Wind Fields over Sloping Terrain*, volume 1452 (1). IOP Publishing, 2020.

M. Sjöholm, T. Mikkelsen, J. Mann, K. Enevoldsen, and M. Courtney. Spatial averaging-effects on turbulence measured by a continuous-wave coherent lidar. *Meteorologische Zeitschrift (Berlin)*, 18, 2009.

R. B. Stull. *An introduction to boundary layer meteorology*, volume 13. Springer Science & Business Media, 1988.

J. Teixeira, B. Stevens, C. Bretherton, R. Cederwall, J. D. Doyle, J.-C. Golaz, A. A. Holtslag, S. Klein, J. K. Lundquist, D. A. Randall, et al. Parameterization of the atmospheric boundary layer: a view from just above the inversion. *Bulletin of the American Meteorological Society*, 89(4):453–458, 2008.

N. Vasiljević, M. Harris, A. Tegtmeier Pedersen, G. Rolighed Thorsen, M. Pitter, J. Harris, K. Bajpai, and M. Courtney. Wind sensing with drone-mounted wind lidars: proof of concept. *Atmospheric Measurement Techniques*, 13(2):521–536, 2020.

J. C. Wyngaard. *Turbulence in the Atmosphere*. Cambridge University Press, 2010.

Y. Zheng, S. Yang, X. Liu, J. Wang, T. Norton, J. Chen, and Y. Tan. The computational fluid dynamic modeling of downwash flow field for a six-rotor uav. *Frontiers of Agricultural Science and Engineering*, 5(2):159–167, 2018.

---

## Author Response (AR2)

Responses to Reviewers' Comments for Manuscript

**Rotary-wing drone-induced flow – comparison of simulations with lidar measurements.**

Addressed Comments for Publication to

Atmospheric Measurement Techniques

by

Liqin Jin, Mauro Ghirardelli, Jakob Mann, Mikael Sjöholm, Stephan T. Kral, and Joachim Reuder

Dear Reviewer,

First of all, we would like to express our gratitude to you for providing your insightful comments on our paper. In this revision, we have carefully incorporated and addressed the two comments that have been raised. This short report summarizes the main modifications we made and a detailed point-by-point response to the comments.

We sincerely appreciate your time and effort in reviewing our manuscript, and we are grateful for the opportunity to address your concerns.

Sincerely,

Liqin Jin, Mauro Ghirardelli, Jakob Mann, Mikael Sjöholm, Stephan T. Kral, and Joachim Reuder

**Authors' Response to Reviewer**

> **General Comments.** Jin et al. have significantly improved the manuscript with regards to language, clarity and scientific soundness. All reviewer comments have been properly addressed and changes were made wherever possible. The problems with the experimental setup could of course not be changed, but I still think the manuscript is worth publishing in AMT after two more comments are addressed.

**Response:**

Dear Reviewer, thank you for your contributions to improve the quality of this manuscript. After reading through your comments, we implemented changes to the introduction and conclusion parts in the manuscript. We will now address, one by one, the two comments.

**Comment 1**

The introduction has been rewritten completely and is in much better shape, but I think a very important part is still missing, which is to describe the state-of-the-art of wind measurements with multicopter UAS in their full extent. The branch of using the avionic information and thus the drone itself as the sensor is not mentioned at all, but is widely used, well validated and shows great potential (Neumann et al. 2015, Segales et al. 2020, Wetz et al. 2021, Wetz and Wildmann 2022, Gonzales-Rocha et al. 2023...). As a direct alternative to the proposed method, this needs to be mentioned in the description of the state-of-the-art. In the discussion, the method with external sonic anemometer should be benchmarked against the uncertainties that are estimated for the wind measurement in those studies.

**Response:**

Thank you for pointing out this valuable comment. We acknowledge the importance of completing the state-of-the-art wind measurements with multicopter UAVs themselves

or with external devices mounted on UAVs. We have expanded our literature review to include the significant contributions of Neumann and Bartholmai [2015b], Segales et al. [2020b], Wetz et al. [2021], Wildmann and Wetz [2022], González-Rocha et al. [2023], which is implemented in the manuscript in **L38** as Since the beginning of the 21st century, uncrewed aerial vehicles (UAVs) with rotary-wings have become more popular for conducting atmospheric measurements [Hemingway et al., 2017, Leuenberger et al., 2020, Tikhomirov et al., 2021], due to their flexibility in orienting, precise hovering capabilities and ease of deployment. Wind velocity and direction can be reconstructed from either the avionic information of UAVs alone [Neumann and Bartholmai, 2015a, Palomaki et al., 2017, Segales et al., 2020a, Wetz et al., 2021, González-Rocha et al., 2023], or from wind sensors mounted on the UAVs. Even though the former indirect approach is well-established and has the advantage of not requiring external measurement devices, it has limitations in resolving three-dimensional wind fields and fine-scale turbulent fluctuations. An exception is found in Wildmann and Wetz [2022], where all three velocity components are measured at a frequency of 1 Hz. However, a sampling frequency of 10 Hz to 20 Hz is typically necessary to resolve the smallest turbulent scales. Therefore, the latter method of using fast-response 3D anemometers, such as sonic anemometers, is inevitable for direct observations of turbulence. This approach extends measurement capabilities, but it may reduce flight performance due to the added weight.

Finally, thank you for your suggestion regarding the benchmark of the proposed method against the uncertainties with the external sonic anemometer. This point will be addressed in the discussion and conclusion part, which is Furthermore, a full-size sonic anemometer could be mounted on the drone in the upstream direction to validate the potentially optimal position defined by CFD simulations and to benchmark wind velocity uncertainties with an external sonic anemometer.

> **Comment 2**
>
> p.22, l.362: "a less stringent threshold": what is the threshold to find the 2-m distance?

**Response:**

Thank you for this comment. We agree with this point and explain what a less stringent threshold is. The new sentence in **L295** is With a less strict threshold of 5% velocity deviation for both horizontal and vertical winds, this 5-meter distance can be substantially brought down to 2 meters when a background flow of at least 4 ms$^{-1}$ is present, corresponding to a flow distortion of $\pm0.2$ ms$^{-1}$ (Fig. 8e). This deviation is similar to the accuracy reported by Wetz et al. [2021] as well as Wildmann and Wetz [2022].

**References**

J. González-Rocha, L. Bilyeu, S. D. Ross, H. Foroutan, S. J. Jacquemin, A. P. Ault, and D. G. Schmale. Sensing atmospheric flows in aquatic environments using a multirotor small uncrewed aircraft system (suas). *Environmental Science: Atmospheres*, 3(2): 305–315, 2023.

B. L. Hemingway, A. E. Frazier, B. R. Elbing, and J. D. Jacob. Vertical sampling scales for atmospheric boundary layer measurements from small unmanned aircraft systems (suas). *Atmosphere*, 8(9):176, 2017.

D. Leuenberger, A. Haefele, N. Omanovic, M. Fengler, G. Martucci, B. Calpini, O. Fuhrer, and A. Rossa. Improving high-impact numerical weather prediction with lidar and drone observations. *Bulletin of the American Meteorological Society*, 101(7):E1036–E1051, 2020.

P. P. Neumann and M. Bartholmai. Real-time wind estimation on a micro unmanned aerial vehicle using its inertial measurement unit. *Sensors and Actuators, A: Physical*, 235:300–310, 11 2015a. ISSN 09244247.

P. P. Neumann and M. Bartholmai. Real-time wind estimation on a micro unmanned aerial vehicle using its inertial measurement unit. *Sensors and Actuators A: Physical*, 235:300–310, 2015b.

R. T. Palomaki, N. T. Rose, M. van den Bossche, T. J. Sherman, and S. F. De Wekker. Wind estimation in the lower atmosphere using multirotor aircraft. *Journal of Atmospheric and Oceanic Technology*, 34(5):1183–1191, 2017.

A. R. Segales, B. R. Greene, T. M. Bell, W. Doyle, J. J. Martin, E. A. Pillar-Little, and P. B. Chilson. The coptersonde: an insight into the development of a smart unmanned aircraft system for atmospheric boundary layer research. *Atmospheric Measurement Techniques*, 13(5):2833–2848, 2020a. doi: 10.5194/amt-13-2833-2020. URL https://amt.copernicus.org/articles/13/2833/2020/.

A. R. Segales, B. R. Greene, T. M. Bell, W. Doyle, J. J. Martin, E. A. Pillar-Little, and P. B. Chilson. The coptersonde: an insight into the development of a smart unmanned

aircraft system for atmospheric boundary layer research. *Atmospheric Measurement Techniques*, 13(5):2833–2848, 2020b.

A. B. Tikhomirov, G. Lesins, and J. R. Drummond. Drone measurements of surface-based winter temperature inversions in the high arctic at eureka. *Atmospheric Measurement Techniques*, 14(11):7123–7145, 2021.

T. Wetz, N. Wildmann, and F. Beyrich. Distributed wind measurements with multiple quadrotor unmanned aerial vehicles in the atmospheric boundary layer. *Atmospheric Measurement Techniques*, 14(5):3795–3814, 2021. doi: 10.5194/amt-14-3795-2021. URL `https://amt.copernicus.org/articles/14/3795/2021/`.

N. Wildmann and T. Wetz. Towards vertical wind and turbulent flux estimation with multicopter uncrewed aircraft systems. *Atmospheric Measurement Techniques*, 15(18): 5465–5477, 2022. doi: 10.5194/amt-15-5465-2022. URL `https://amt.copernicus.org/articles/15/5465/2022/`.